# Stateful ODE-Nets using Basis Function Expansions

**Alejandro Queiruga**[*]
Google Research
afq@google.com

**N. Benjamin Erichson**[*]
University of Pittsburgh
erichson@pitt.edu

**Liam Hodgkinson**
ICSI and UC Berkeley
liam.hodgkinson@berkeley.edu

**Michael W. Mahoney**
ICSI and UC Berkeley
mmahoney@stat.berkeley.edu

## Abstract

The recently-introduced class of ordinary differential equation networks (ODE-Nets) establishes a fruitful connection between deep learning and dynamical systems. In this work, we reconsider formulations of the weights as continuous-in-depth functions using linear combinations of basis functions which enables us to leverage parameter transformations such as function projections. In turn, this view allows us to formulate a novel stateful ODE-Block that handles stateful layers. The benefits of this new ODE-Block are twofold: first, it enables incorporating meaningful continuous-in-depth batch normalization layers to achieve state-of-the-art performance; second, it enables compressing the weights through a change of basis, without retraining, while maintaining near state-of-the-art performance and reducing both inference time and memory footprint. Performance is demonstrated by applying our stateful ODE-Block to (a) image classification tasks using convolutional units and (b) sentence-tagging tasks using transformer encoder units.

## 1 Introduction

The interpretation of neural networks (NNs) as discretizations of differential equations [7, 17, 31, 43] has recently unlocked a fruitful link between deep learning and dynamical systems. The strengths of so-called ordinary differential equation networks (ODE-Nets) are that they are well suited for modeling time series [11, 39] and smooth density estimation [14]. They are also able to learn representations that preserve the topology of the input space [9] (which may be seen as a "feature" or as a "bug"). Further, they can be designed to be highly memory efficient [13, 51]. However, one major drawback of current ODE-Nets is that the predictive accuracy for tasks such as image classification is often inferior as compared to other state-of-the-art NN architectures. The reason for the poor performance is two-fold: (a) ODE-Nets have shallow parameterizations (albeit long computational graphs), and (b) ODE-Nets do not include a mechanism for handling layers with internal state (i.e., stateful layers, or stateful modules), and thus cannot leverage batch normalization layers which are standard in image classification problems. That is, in modern deep learning environments the running mean and variance statistics of a batch normalization module are not trainable parameters, but are instead part of the module's state, which does not fit into the ODE framework.

Further, traditional techniques such as stochastic depth [21], layer dropout [12] and adaptive depth [10, 28], which are useful for regularization and compressing traditional NNs, do not necessarily apply in a meaningful manner to ODE-Nets. That is, these methods do not provide a systematic scheme to derive smaller networks from a single deep ODE-Net source model. This limitation prohibits quick and rigorous adaptation of current ODE-Nets across different computational environments.

---

[*]Equal contributions.

35th Conference on Neural Information Processing Systems (NeurIPS 2021).

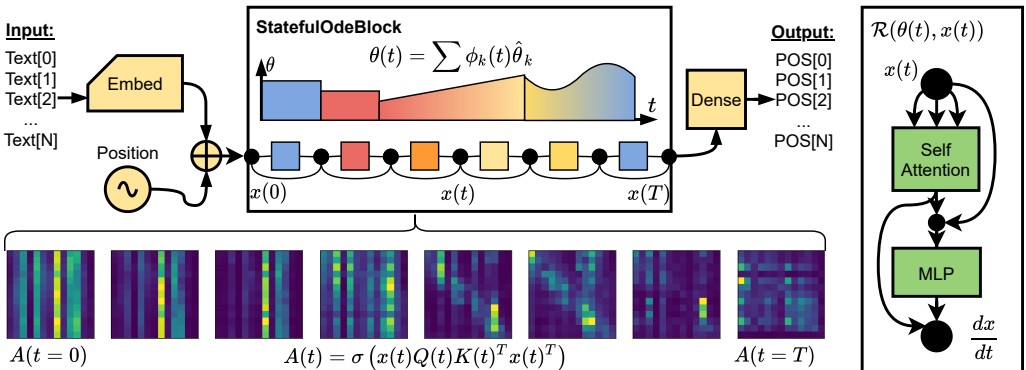

Figure 1: Sketch of a continuous-in-depth transformer-encoder. The model architecture consists of a sparse embedding layer followed by an OdeBlock that integrates the encoder to feed into a classification layer to determine parts of speech. The model graph is generated by nesting the residual $\mathcal{R}$ on the right into a time-integration scheme in the OdeBlock. The weights for each call to $\mathcal{R}$ are determined by evaluating the basis expansion. Internal hidden states evolve continuously during the forward pass. This is illustrated by a smoothly varying attention matrix.

Hence, there is a need for a general but effective way of reducing the number of parameters of deep ODE-Nets to reduce both inference time and the memory footprint.

To address these limitations, we propose a novel stateful ODE-Net model that is built upon the theoretical underpinnings of numerical analysis. Following recent work by [34, 37], we express the weights of an ODE-Net as continuous-in-depth functions using linear combinations of basis functions. However, unlike prior works, we consider parameter transformations through a change of basis functions, i.e., *adding basis functions*, *decreasing the number of basis functions*, and *function projections*. In turn, we are able to (a) design deep ODE-Nets that leverage meaningful continuous-in-depth batch normalization layers to boost the predictive accuracy of ODE-Nets, e.g., we achieve 94.4% test accuracy on CIFAR-10, and 79.9% on CIFAR-100; and (b) we introduce a methodology for compressing ODE-Nets, e.g, we are able to reduce the number of parameters by a factor of two, *without retraining or revisiting data*, while nearly maintaining the accuracy of the source model.

In the following, we refer to our model as *Stateful ODE-Net*. Here are our main contributions.

- **Stateful ODE-Block.** We introduce a novel stateful ODE-Block that enables the integration of stateful network layers (Sec. 4.1). To do so, we view continuous-in-depth weight functions through the lens of basis function expansions and leverage basis transformations. An example of our continuous-in-depth model is shown in Figure 1.

- **Stateful Normalization Layer.** We introduce stateful modules using function projections. This enables us to introduce continuous-in-depth batch normalization layers for ODE-Nets (Sec. 4.2). In our experiments, we show that such stateful normalization layers are crucial for ODE-Nets to achieve state-of-the-art performance on mainstream image classification problems (Sec. 7).

- *A Posteriori* **Compression Methodology.** We introduce a methodology to compress ODE-Nets without retraining or revisiting any data, based on parameter interpolation and projection (Sec. 5). That is, we can systematically derive smaller networks from a single deep ODE-Net source model through a change of basis. We demonstrate the accuracy and compression performance for various image classification tasks using both shallow and deep ODE-Nets (Sec. 7).

- **Advantages of Higher-order Integrators for Compression.** We examine the effects of training continuous-in-depth weight functions through their discretizations (Sec. 6). Our key insight in Theorem 1 is that higher-order integrators introduce additional implicit regularization which is crucial for obtaining good compression performance in practice. Proofs are provided in Appendix A.

## 2 Related Work

The "formulate in continuous time and then discretize" approach [7, 17] has recently attracted attention both in the machine learning and applied dynamical systems community. This approach

considers a differential equation that defines the continuous evolution from an initial condition $x(0) = x_{in}$ to the final output $x(T) = x_{out}$ as

$$\dot{x}(t) = \mathcal{F}(\hat{\theta}, x(t), t). \tag{1}$$

Here, the function $\mathcal{F}$ can be any NN that is parameterized by $\hat{\theta}$, with two inputs $x$ and $t$. The parameter $t \in [0, T]$ in this ODE represents time, analogous to the depth of classical network architectures. Using a finer temporal discretization (with a smaller $\Delta t$) or a larger terminal time $T$, corresponds to deeper network architectures. Feed-forward evaluation of the network is performed by numerically integrating Eq. (1):

$$x_{out} = x_{in} + \int_0^T \mathcal{F}(\hat{\theta}, x(t), t) \, \mathrm{d}t = \texttt{OdeBlock}\left[\mathcal{F}, \hat{\theta}, \texttt{scheme}, \Delta t, t \in [0, T]\right](x_{in}). \tag{2}$$

Inspired by this view, numerous ODE and PDE-based network architectures [1, 8, 9, 15, 32, 42, 48, 49], and continuous-time recurrent units [4, 29, 30, 40, 41] have been proposed.

Recently, the idea of representing a continuous-time weight function as a linear combination of basis functions has been proposed by [34] and [37]. This involves using the following formulation:

$$\dot{x}(t) = \mathcal{R}(\theta(t; \hat{\theta}), x(t)), \tag{3}$$

where $\mathcal{R}$ is now parameterized by a continuously-varying weight function $\theta(t; \hat{\theta})$. In turn, this weight function is parameterized by a countable tensor of trainable parameters $\hat{\theta}$. Both of these works noted that piecewise constant basis functions algebraically resemble residual networks and stacked ODE-Nets. A similar concept was used by [6] to inspire a multi-level refinement training scheme for discrete ResNets. In addition, [34] uses orthogonal basis sets to formulate a Galërkin Neural ODE.

In this work, we take advantage of basis transformations to introduce stateful normalization layers as well as a methodology for compressing ODE-Nets, thus improving on ContinuousNet and other prior work [37]. Although basis elements are often chosen to be orthogonal, inspired by multi-level refinement training, we shall consider non-orthogonal basis sets in our experiments. Table 1 highlights the advantages of our model compared to other related models.

Table 1: Comparison of our Stateful ODE-Net to other dynamical system inspired models.

| Model | Multi-level | Compression | Basis Function View | Stateful Layers |
|---|---|---|---|---|
| NODE [7] | ✗ | ✗ | ✗ | ✗ |
| Multi-level ResNet [6] | ✓ | ✗ | ✗ | ✗ |
| Galerkin ODE-Net [34] | ✗ | ✗ | ✓ | ✗ |
| ContinuousNet [37] | ✓ | ✗ | ✓ | ✗ |
| Stateful ODE-Net (ours) | ✓ | ✓ | ✓ | ✓ |

## 3 Basis Function View of Continuous-in-depth Weight Functions

Let $\theta(t; \hat{\theta})$ be an arbitrary weight function that depends on depth (or time) $t$, which takes as argument a vector of real-valued weights $\hat{\theta}$. Given a basis $\phi$, we can represent $\theta$ as a linear combination of $K$ (continuous-time) basis functions $\phi_k(t)$:

$$\theta(t; \hat{\theta}) = \sum_{k=1}^{K} \phi_k(t) \, \hat{\theta}_k. \tag{4}$$

The basis sets which we consider have two parameters that specify the family of functions $\phi$ and cardinality $K$ (i.e., the number of functions) to be used. Hence, we represent a basis set by $(\phi, K)$. While our methodology can be applied to any basis set, we restrict our attention to piecewise constant and piecewise linear basis functions, common in finite volume and finite element methods [44, §4].

- **Piecewise constant basis.** This orthogonal basis consist of basis functions that assume a constant value for each of the elements of width $\Delta t = T/K$, where $T$ is the "time" at the end of the ODE-Net. The summation in Eq. (4) involves piecewise-constant indicator functions $\phi_k(t)$ satisfying

$$\phi_k(t) = \begin{cases} 1, & t \in [(k-1)\Delta t, \, k\Delta t] \\ 0, & \text{otherwise.} \end{cases} \tag{5}$$

- **Piecewise linear basis functions.** This basis consists of evenly spaced elements where each parameter $\hat{\theta}_k$ corresponds to the value of $\theta(t_k)$ at element boundaries $t_k = T(k-1)/(K-1)$. Each basis function is a "hat" function around the point $t_k$,

$$\phi_k(t) = \begin{cases} (t - k\Delta t)/\Delta t, & t \in [(k-1)\Delta t, \, k\Delta t] \\ 1 - (t - k\Delta t)/\Delta t, & t \in [k\Delta t, \, (k+1)\Delta t] \\ 0, & \text{otherwise.} \end{cases} \tag{6}$$

Piecewise linear basis functions have compact and overlapping supports; i.e., they are not orthogonal, unlike piecewise constant basis functions.

### 3.1 Basis Transformations

We consider two avenues of basis transformation to change function representation: interpolation and projection. Note that these transformations will often introduce approximation error, particularly when transforming to a basis with a smaller size. Furthermore, interpolation and projection do not necessarily give the same result for the same function.

**Interpolation.** Some basis functions use control points for which parameters correspond to values at different $t_k$ such that $\theta(t_k, \hat{\theta}) = \hat{\theta}_k$. Given $\theta^1$, the parameter coefficients for $\theta^2$ can thus be calculated by evaluating the $\theta^1$ at the control points $t_k^2$:

$$\hat{\theta}_k^2 = \theta^1(t_k^2, \hat{\theta}^1) = \sum_{b=1}^{K_1} \phi_b^1(t_k^2)\hat{\theta}_b^1 \quad \text{for } k = 1, ..., K_2. \tag{7}$$

Interpolation only works with basis functions where the parameters correspond to control point locations. For piecewise constant basis functions, $t_k$ corresponds to the cell centers, and for piecewise linear basis functions, $t_k$ corresponds to the endpoints at element boundaries.

**Projection.** Function projection can be used with any basis expansion. Given the function $\theta^1(t)$, the coefficients $\hat{\theta}_k^2$ are solved by a minimization problem:

$$\min_{\hat{\theta}_k^2} \int_0^T \left( \theta^1(t, \hat{\theta}^1) - \theta^2(t, \hat{\theta}^2) \right)^2 \mathrm{d}t = \min_{\hat{\theta}_k^2} \int_0^T \left( \sum_{a=1}^{K_1} \hat{\theta}_a^1 \phi_a^1(t) - \sum_{k=1}^{K_2} \hat{\theta}_k^2 \phi_k^2(t) \right)^2 \mathrm{d}t. \tag{8}$$

Appendix C.1 includes the details of the numerical approximation of the loss and its solution. The integral is evaluated using Gaussian quadrature over sub-cells. The overall calculation solves the same linear problem repeatedly for each coordinate of $\theta$ used by the call to $\mathcal{R}$.

## 4 Stateful ODE-Nets

Modern neural network architectures leverage stateful modules such as BatchNorms [22] to achieve state-of-the-art performance, and recent work has demonstrated that ODE-Nets also can benefit from normalization layers [16]. However, incorporating normalization layers into ODE-Nets is challenging; indeed, recent work [45] acknowledges that it was not obvious how to include BatchNorm layers in ODE-Nets. The reason for this challenge is that normalization layers have internal state parameters with specific update rules, in addition to trainable parameters.

### 4.1 Stateful ODE-Block

To formulate a stateful ODE-Block, we consider the following differential equation:

$$\dot{x}(t) = \mathcal{R}\left( \theta^g(t), \theta^s(t), x(t) \right), \tag{9}$$

which is parameterized by two continuous-in-depth weight functions $\theta^g(t, \hat{\theta}^g)$ and $\theta^s(t, \hat{\theta}^s)$, respectively. For example, the continuous-in-depth BatchNorm function has two gradient-parameter functions in $\theta^g(t)$ – scale $s(t)$ and bias $b(t)$ – and two state-parameter functions in $\theta^s(t)$ – mean $\mu(t)$ and variance $\sigma(t)$. Using a functional programming paradigm inspired by Flax [20], we replace the

internal state update of the module with a secondary output. The continuous update function $\mathcal{R}$ is split into two components: the forward-pass output $\dot{x} = \mathcal{R}_x$ and the state update output $\theta^{s*} = \mathcal{R}_s$. Then, we solve $\dot{x}$ using methods for numerical integration (denoted by `scheme`), followed by a basis projection of $\hat{\theta}^{s*}$. We obtain the following input-output relation for the forward pass during training:

$$\begin{cases} x_{out} & = \texttt{StatefulOdeBlock}_x \left[ \mathcal{R}_x, \texttt{scheme}, \Delta t, \phi \right] (\hat{\theta}^g, \hat{\theta}^s, x_{in}) \\ \hat{\theta}^{s*} & = \texttt{StatefulOdeBlock}_s \left[ \mathcal{R}_s, \texttt{scheme}, \Delta t, \phi \right] (\hat{\theta}^g, \hat{\theta}^s, x_{in}), \end{cases} \quad (10)$$

which we jointly optimize for $\theta^g(t)$ and $\theta^s(t)$ during training. Optimizing for $\theta^g$ and $\theta^s$ involves two coupled equations, updating the gradient with respect to the loss $L$ and a fixed-point iteration on $\theta^s$,

$$\theta^{g*}(t) = \theta^g(t) - \tfrac{\partial L}{\partial \theta^g(t)}(\theta^g(t), \theta^s(t), x_0), \quad (11)$$

$$\theta^{s*}(t) = \mathcal{R}_s\left(\theta^g(t), \theta^s(t), x(t)\right). \quad (12)$$

The updates to $\theta^g(t)$ are computed by backpropagation of the loss through the ODE-Block with respect to its basis coefficients $\frac{\partial x_{out}}{\partial \hat{\theta}_g}$. Optimizing $\hat{\theta}^s(t)$ involves projecting the update rule back onto the basis during every training step.

## 4.2 Numerical Solution of Stateful Normalization Layers

While Eq. (12) can be computed given a forward pass solution $x(t)$, this naive approach requires implementing an additional numerical discretization. Instead, consider that each call to $\mathcal{R}$ at times $t_i \in [0, T]$ during the forward pass outputs an updated state $\bar{\theta}_i^s = \mathcal{R}_s(t_i)$. This generates a point cloud $\{t_i, \bar{\theta}_i^s\}$ which can be projected back onto the basis set by minimizing the point-wise least-squared-error

$$\hat{\theta}^{s*} = \arg\min_{\hat{\theta}^{s*}} \sum_{i=1}^{N_{state}} \left( \bar{\theta}_i^s - \sum_{k=1}^{K} \phi_k(t_i) \hat{\theta}_k^{s*} \right)^2. \quad (13)$$

Algorithm 1 in the appendix describes the calculation of Eq. (10), fused into one loop. When using forward Euler integration and piecewise constant basis functions, this algorithm reduces to the forward pass and update rule in a ResNet with BatchNorm layers. This theoretical formalism and algorithm generalizes to any combination of stateful layer, basis set, and integration scheme.

## 5 *A Posteriori* Compression Methodology

Basis transformations of continuous functions are a natural choice for compressing ODE-Nets because they couple well with continuous-time operations. Interpolation and projection can be applied to the coefficients $\hat{\theta}^g$ and $\hat{\theta}^s$ to change the number of parameters needed to represent the continuous functions. Given the coefficients $\hat{\theta}^1$ to a function $\theta^1(t)$ on a basis $(\phi^1, K_1)$, we can determine new coefficients $\hat{\theta}^2$ on a different space $(\phi^2, K_2)$. Changing the basis size can reduce the total number of parameters (to $K_2 < K_1$), and hence the total storage requirement. This representation enables compression in a systematic way — in particular, we can consider a change of basis to transform

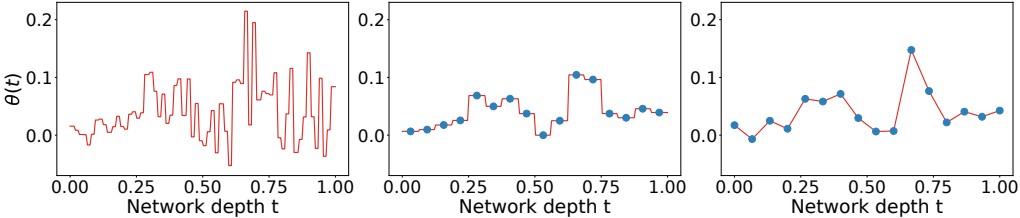

Figure 2: Example of projecting a component of the query kernel from a continuous-in-depth transformer. The model was trained with $K = 64$ piecewise constant basis functions (left) and then projected to: $K = 16$ piecewise constant basis functions (middle), and $K = 16$ piecewise linear basis functions (left). Circles denote the control points (knots) corresponding to parameters $\hat{\theta}_k$.

learned parameter coefficients $\hat{\theta}^1$ to other coefficients $\hat{\theta}^2$. To illustrate this, Figure 2 shows different basis representations of a weight function.

Note that this compression methodology also applies to basis function ODE-Nets that are trained without stateful normalization layers. Further, continuous-time formulations can also decrease the number of model steps, increasing inference speed, in addition to decreasing the model size.

# 6   Advantages of Higher-order Integrators for Compression

To implement any ODE-Net, it is necessary to approximately solve the corresponding ODE using a numerical integrator. There are many possible integrators one can use, so to take full advantage of our proposed methodology, we examine the advantages of certain integrators on compression from a theoretical point of view. In short, for the same step size, higher-order integrators exhibit increasing numerical instability if the underlying solution changes too rapidly. Because these large errors will propagate through the loss function, minimizing any loss computed on the numerically integrated ODE-Net will avoid choices of the weights that result in rapid dynamics. This is beneficial for compression, as slower, smoother dynamics in the ODE allow for coarser approximations in $\theta$.

To quantify this effect, we derive an asymptotic global error term for small step sizes. Our analysis focuses on *explicit Runge–Kutta (RK) methods*, although similar treatments are possible for most other integrators. For brevity, here we use $h$ to denote step size in place of $\Delta t$. A $p$-stage explicit RK method for an ODE $\dot{y}_\theta(t) = f(y_\theta(t), \theta(t))$ provides an approximation $y_{h,\theta}(t)$ of $y_\theta(t)$ for a given step size $h$ on a discrete grid of points $\{0, h, 2h, \dots\}$ (which can then be joined together through linear interpolation). As the order $p$ increases, the error in the RK approximation generally decreases more rapidly in the step size for smooth integrands. However, a tradeoff arises through an increased sensitivity to any irregularities. We consider these properties from the perspective of *implicit regularization*, where the choice of integrator impacts the trained parameters $\theta$.

Implicit regularization is of interest in machine learning very generally [33], and in recent years it has received attention for NNs [25, 35]. In a typical theory-centric approach to describing implicit regularization, one derives an approximately equivalent *explicit regularizer*, relative to a more familiar loss function. In Theorem 1, we demonstrate that for any scalar loss function $L$, using a RK integrator *implicitly regularizes* towards smaller derivatives in $f$. Since $\theta$ can be arbitrary, we consider finite differences in time. To this effect, recall that the $m$-th order forward differences are defined by $\Delta_h^m f(x, t) = \Delta_h^{m-1} f(x, t+h) - \Delta_h^{m-1} f(x, t)$, with $\Delta_h^0 f(x, t) = f(x, t)$. Also, for any $t \in [0, T]$, we let $\iota_h(t) = \lfloor t/h \rfloor \cdot h$ denote the nearest point on the grid $\{0, h, 2h, \dots\}$ to $t$.

**Theorem 1 (Implicit regularization)** *There exists a polynomial $P$ depending only on the Runge-Kutta scheme satisfying $P(0) = 0$, and a smooth function $\bar{y}_\theta(t)$ depending on $f, \theta$, and the scheme, such that for any $t \in [0, T]$, as $h \to 0^+$,*

$$L(y_{h,\theta}(t)) = L(\bar{y}_\theta(t)) + h^p \nabla L(\bar{y}_\theta(t)) \cdot e_{h,\theta}(t) + \mathcal{O}(h^{p+1}), \tag{14}$$

*where $\dot{e}_{h,\theta}(t) = \frac{\partial f}{\partial y}(y_{h,\theta}(t), \theta(\iota_h(t))) e_{h,\theta}(t) + P \circ \mathcal{D}_{h,\theta}^p(t)$ and*

$$\mathcal{D}_{h,\theta}^p(t) = \left\{ h^{-m} \Delta_h^m \frac{\partial^l f_j}{\partial y^l} f_j(y_{h,\theta}(t), \theta(\iota_h(t))) \right\}_{j=1,\dots,d,\, l+m \leq p+1}. \tag{15}$$

The proof is provided in Appendix A. Theorem 1 demonstrates that Runge-Kutta integrators implicitly regularize toward smaller derivatives/finite differences of $f$ of orders $k \leq p + 1$. To demonstrate the effect this has on compression, we consider the *sensitivity* of the solution $y$ to $\theta(t)$, and hence to the choice of basis functions, using Gateaux derivatives. A smaller sensitivity to $\theta(t)$ allows for coarser approximations before encountering a significant reduction in accuracy. The sensitivity of the solution in $\theta(t)$ is contained in the following lemma.

**Lemma 1** *There exists a smooth function $\bar{\theta}$ depending only on $\theta$ such that for any smooth function $\varphi(t)$,*

$$D_\varphi y_{h,\theta}(t) := \frac{\mathrm{d}}{\mathrm{d}\epsilon} y_{h,\theta+\epsilon\varphi}(t) \bigg|_{\epsilon=0} = \int_0^t e^{F_{h,\theta}(s,t)} \frac{\partial f}{\partial \theta}(y_{h,\theta}(s), \bar{\theta}(s)) \varphi(s) \mathrm{d}s + \mathcal{O}(h^p),$$

*where $F_{h,\theta}(s,t) = \int_s^t \frac{\partial f}{\partial y}(y_{h,\theta}(u), \bar{\theta}(u)) \mathrm{d}u$.*

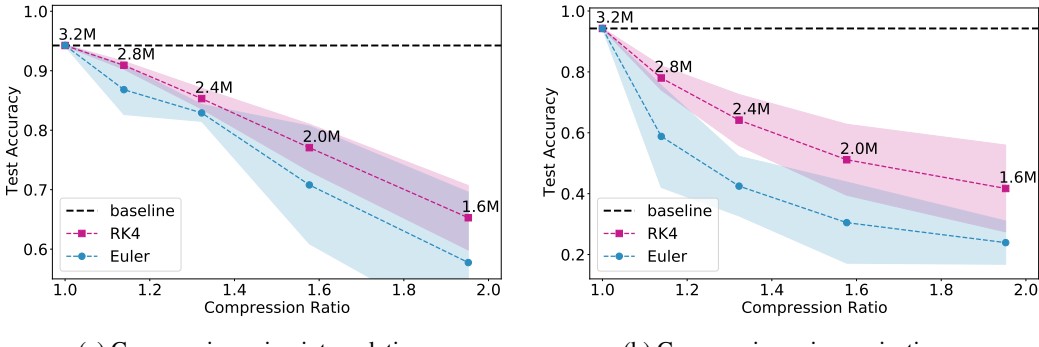

(a) Compression using interpolation.    (b) Compression using projections.

Figure 3: Higher-order integrators introduce additional implicit regularization into learned continuous weight functions. By experimenting on CIFAR-10, it is empirically observed that training with the RK4 scheme improves the compression-performance compared to using the forward Euler scheme.

Simply put, Lemma 1 shows that the sensitivity of $y_{h,\theta}(t)$ in $\theta$ decreases monotonically with decreasing derivatives of the integrand $f$ in each of its arguments. These are precisely the objects that appear in Theorem 1. Therefore, a reduced sensitivity can occur in one of two ways:

(I) A higher-order integrator is used, causing higher-order derivatives to appear in the implicit regularization term $e_{h,\theta}$. By the Landau-Kolmogorov inequality [26], for any differential $D$ and integer $m \geq 1$, $\|D^m f\| \geq c_m \|f\| (\|Df\|/\|f\|)^m$ for some $c_m > 0$. Hence, by implicitly regularizing towards smaller higher-order derivatives/finite differences, we should expect more dramatic reductions in the first-order derivative/finite difference as well.

(II) A larger step size $h$ is used. Since doing so can lead to stability concerns during training, we later consider a refinement training scheme [6, 37] where $h$ is slowly reduced.

In Figure 3, we verify strategy (I), showing that the 4th-order RK4 scheme exhibits improved test accuracy for higher compression ratios over the 1st-order Euler scheme. Unfortunately, higher-order integrators typically increase the runtime on the order of $\mathcal{O}(p)$. Therefore, some of our later experiments will focus on strategy (II), which avoids this issue.

## 7  Empirical results

We present empirical results to demonstrate the predictive accuracy and compression performance of Stateful ODE-Nets for both image classification and NLP tasks. Each experiment was repeated with eight different seeds, and the figures report mean and standard deviations. Details about the training process, and different model configurations are provided in Appendix E. Research code is provided as part of the following GitHub repository: `https://github.com/afqueiruga/StatefulOdeNets`. Our models are implemented in Jax [3], using Flax [20].

### 7.1  Compressing Shallow ODE-Nets for Image Classification Tasks

First, we consider shallow ODE-Nets, which have low parameter counts, in order to compare our models to meaningful baselines from the literature. We evaluate the performance on both MNIST [27] and CIFAR-10 [24]. Here, we train our Stateful ODE-Net using the classic Runge-Kutta (RK4) scheme and the multi-level refinement method proposed by [6, 37].

**Results for MNIST.** Due to the simplicity of this problem, we consider for this experiment a model that has only a single OdeBlock with 8 units, and each unit has 12 channels. Our model has about $18K$ parameters and achieves about $99.4\%$ test accuracy on average. Despite the lower parameter count, we outperform all other ODE-based networks on this task, as shown in Table 2. We also show an ablation model by training our ODE-Net without continuous-in-depth batch normalization layers while keeping everything else fixed. The ablation experiment shows that normalization provides only a slight accuracy improvements; this is to be expected as MNIST is a relatively easy problem.

Table 2: Compression performance and test accuracy of shallow ODE-Nets on MNIST.

| Model | Best | Average | Min | # Parameters | Compression | Inference |
|---|---|---|---|---|---|---|
| NODE [9] | - | 96.4% | - | 84K | - | - |
| ANODE [9] | - | 98.2% | - | 84K | - | - |
| 2nd-Order [34] | - | 99.2% | - | 20K | - | - |
| A4+NODE+NDDE [50] | - | 98.5% | - | 84K | - | - |
| Ablation Model | 99.3% | 99.1% | 98.9% | 18K | - | - |
| Stateful ODE-Net (ours) | **99.6%** | **99.4%** | **99.3%** | 18K | baseline | 1.7 (s) |
| ↪ (compressed) | 99.4% | 99.2% | 99.1% | 10K | 45% | 1.2 (s) |
| ↪ (compressed) | 97.8% | 96.9% | 95.7% | 7K | 61% | 1.1 (s) |

Table 3: Compression performance and test accuracy of shallow ODE-Nets on CIFAR-10.

| Model | Best | Average | Min | # Parameters | Compression | Inference |
|---|---|---|---|---|---|---|
| HyperODENet [45] | - | 87.9% | - | 460K | - | - |
| SDE BNN (+ STL) [45] | - | 89.1% | - | 460K | - | - |
| Hamiltonian [42] | - | 89.3% | - | 264K | - | - |
| NODE [9] | - | 53.7% | - | 172K | - | - |
| ANODE [9] | - | 60.6% | - | 172K | - | - |
| A4+NODE+NDDE [50] | - | 59.9% | - | 107K | - | - |
| Ablation Model | 88.9% | 88.4% | 88.1% | 204K | - | - |
| Stateful ODE-Net (ours) | **90.7%** | **90.4%** | **90.1%** | 207K | baseline | 2.1 (s) |
| ↪ (compressed) | 90.3% | 89.9% | 89.6% | 114K | 45% | 1.6 (s) |

Next, we compress our model by reducing the number of basis functions and timesteps from 8 down to 4. We do this without retraining, fine-tuning, or revisiting any data. The resulting model has approximately $10K$ parameters (a $45\%$ reduction), while achieving about $99.2\%$ test accuracy. We can compress the model even further to $7K$ parameters, if we are willing to accept a $2.6\%$ drop in accuracy. Despite this drop, we still outperform a simple NODE model with $84K$ parameters. Further, we can see that the inference time (evaluated over the whole test set) is significantly reduced.

**Results for CIFAR-10.** Next, we demonstrate the compression performance on CIFAR-10. To establish a baseline that is comparable to other ODE-Nets, we consider a model that has two OdeBlocks with 8 units, and the units in the first block have 16 channels, while the units in the second block have 32 channels. Our model has about $207K$ parameters and achieves about $90.4\%$ accuracy. Note, that our model has a similar test accuracy to that of a ResNet-20 with $270K$ parameters (a ResNet-20 yields about $91.25\%$ accuracy [19]). In Table 3 we see that our model outperforms all other ODE-Nets on average, while having a comparable parameter count. The ablation model is not able to achieve state-of-the-art performance here, indicating the importance of normalization layers.

As before, we compress our model by reducing the number of basis functions and timesteps from 8 down to 4. The resulting model has $114K$ parameters and achieves about $89.9\%$ accuracy. Despite the compression ratio of nearly 2, the performance of our compressed model is still better as compared to other ODE-Nets. Again, it can be seen that the inference time on the test set is greatly reduced.

### 7.2   Compressing Deep ODE-Nets for Image Classification Tasks

Next, we demonstrate that we can also compress high-capacity deep ODE-Nets trained on CIFAR-10. We consider a model that has 3 stateful ODE-blocks. The units within the 3 blocks have an increasing number of channels: 16,32,64. Additional results for CIFAR-100 are provided in Appendix D.

Table 4 shows results for CIFAR-10. Here we consider two configurations: (c1) is a model trained with refinement training, which has piecewise linear basis functions; (c2) is a model trained without refinement training, which has piecewise constant basis functions. It can be seen that model (c2) achieves high predictive accuracy, yet it yields a poor compression performance. In contrast, model (c1) is about $1.5\%$ less accurate, but it shows to be highly compressible. We can compress the param-

Table 4: Compression performance and test accuracy of deep ODE-Nets on CIFAR-10.

| Model | Best | Average | Min | # Parameters | Compression | Inference |
|---|---|---|---|---|---|---|
| ResNet-110 [19] | - | 93.4% | - | 1.73M | - | - |
| ResNet-122-i [6] | - | 93.8% | - | 1.92M | - | - |
| MidPoint-62 [5] | - | 92.8% | - | 1.78M | - | - |
| ContinuousNet [37] | 94.0% | 93.8% | 93.5% | 3.19M | - | - |
| Ablation Model | 10.0% | 10.0% | 10.0% | 1.62K | - | - |
| Stateful ODE-Net (c1) | 93.0% | 92.4% | 92.1% | 1.63M | baseline | 3.4 (s) |
| ↪ (compressed) | 92.2% | 91.8% | 91.1% | 0.85M | 48% | 2.3 (s) |
| Stateful ODE-Net (c2) | **94.4%** | **94.1%** | **93.8%** | 1.63M | baseline | 3.4 (s) |
| ↪ (compressed) | 69.9% | 60.5% | 52.7% | 0.85M | 48% | 2.3 (s) |

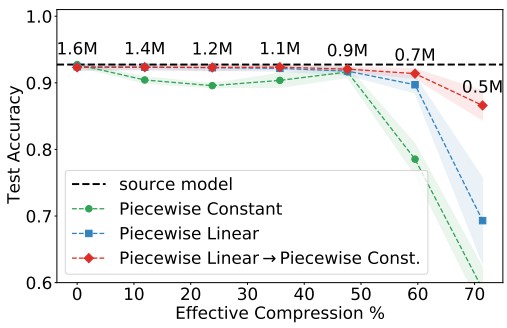

(a) Compressing basis coefficients with fixed $N_T$.

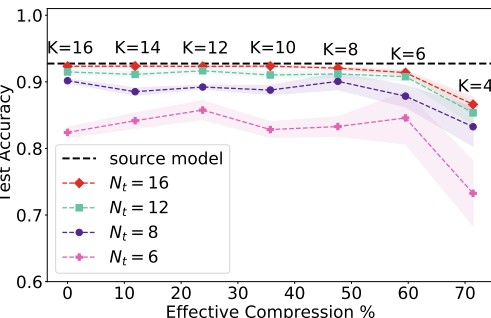

(b) Decreasing $N_T$ using the red line in (a).

Figure 4: In (a) we show the prediction accuracy on CIFAR-10 as a function of weight compression for models that are trained with different basis functions. It can be seen that there is advantage of using piecewise linear basis functions during training. In (b) we compare models with different numbers of time steps. Reducing the number of time-steps reduces the the number of FLOPs.

eters by nearly a factor of 2, while increasing the test error only by $0.6\%$. Further, the ablation model shows that normalization layers are crucial for training deep ODE-Nets on challenging problems. Here, the ablation model that is trained without our continuous-in-depth batch normalization layers does not converge and achieves a performance similar to tossing a 10-sided dice.

In Figure 4a, we show that the compression performance of model (c1) depends on the particular choice of basis set that is used during training and inference time. Using piecewise constant basis functions during training yields models that are slightly less compressible (green line), as compared to models that are using piecewise linear basis functions (blue line). Interestingly, the performance can be even further improved by projecting the piecewise linear basis functions onto piecewise constant basis functions during inference time (red line). In Figure 4b, we show that we can also decrease the number of time steps $N_T$ during inference time, which in turn leads to a reduction of the number of FLOPs during inference time. Recall, $N_T$ refers to the number of time steps, which in turn determines the depth of the model graph. For instance, reducing $N_T$ from 16 to 8, reduces the inference time by about a factor of 2, while nearly maintaining the predictive accuracy.

### 7.3 Compressing Continuous Transformers

A discrete transformer-based encoder can be written as

$$x_{t+1} = x_t + T(q, x_t) + M(\rho, x_t + T(q, x_t)) \tag{16}$$

where $T(q, x)$ is self attention (SA) with parameters $q$ (appearing twice due to the internal skip connection) and $M$ is a multi-layer perceptron (MLP) with parameters $\rho$. Repeated transformer-based encoder blocks can be phrased as an ODE with the equation

$$\dot{x} = T(q(t), x) + M(\rho(t), x + T(q(t), x)). \tag{17}$$

Recognizing $\theta(t) = \{q(t), \rho(t)\}$, this formula can be directly plugged into the basis-functions and OdeBlocks to create a continuous-in-depth transformer, illustrated in Figure 1. Note that the

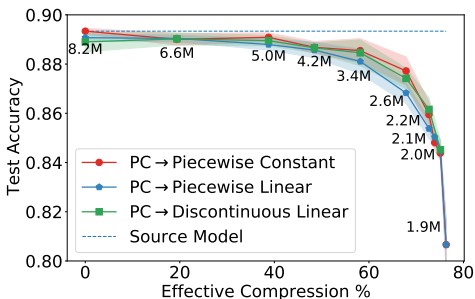

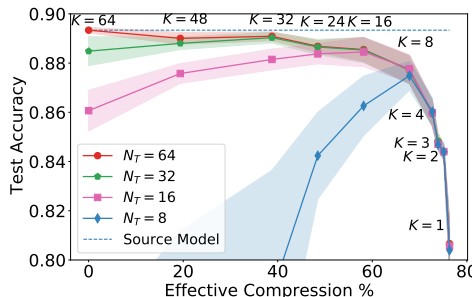

(a) Compressing basis coefficients with fixed $N_T$.

(b) Decreasing $N_T$ using the red line in (a).

Figure 5: In (a) we show the accuracy of a transformer on a part-of-speech tagging problem as a function of weight compression. The transformer model can be compressed by a factor of 2, while nearly maintaining its predictive accuracy. Discounting the 1.8M embedding parameters, the smallest model achieves 98% compression, albeit losing accuracy. In (b) we show models with different number of time steps $N_t$ (i.e., models with with shorter computational graphs).

OdeBlock is continuous along the *depth* of the computation; the sequence still uses discrete tokens and positional embeddings. With a forward Euler integrator, $\Delta t = 1$, a piecewise constant basis, and $K = N_T$, Eq. (17) generates an algebraically equivalent graph to the discrete transformer.

We apply the encoder model to a part-of-speech tagging task, using the English-GUM treebank [47] from the Universal Dependencies treebanks [36]. Our model uses an embedding size of 128, with Key/Query/Value and MLP dimensions of 128 with a single attention head. The final OdeBlock has $K = 64$ piecewise constant basis functions and takes $N_T = 64$ steps.

In Figure 5a, we present the compression performance for different basis sets. Staying on piecewise constant basis functions yields the best performance during testing (red line). The performance slightly drops when we project the piecewise constant basis onto a piecewise linear basis (blue line). Projecting to a discontinuous linear basis (green line) performs approximately as well as the piecewise constant basis. In Figure 5b, we show the effect of decreasing the number of time steps $N_T$ during inference time on the predictive accuracy. In the regime where $N_T \leq K$, there is no significant loss obtained by reducing $K$. Note that there is a divergence when $N_T > K$, where the integration method will skip over parameters. However, projection to $N_T \leq K$ incorporates information across a larger depth-window. Thus, projection improves graph shortening as compared to previous ODE-Nets.

## 8 Conclusion

We introduced a Stateful ODE-based model that can be compressed without retraining, and which can thus quickly and seamlessly be adopted to different computational environments. This is achieved by formulating the weight function of our model as linear combinations of basis functions, which in turn allows us to take advantage of parameter transformations. In addition, this formulation also allows us to implement meaningful stateful normalization layers that are crucial for achieving predictive accuracy comparable to ResNets. Indeed, our experiments showcase that Stateful ODE-Nets outperform other recently proposed ODE-Nets, achieving $94.1\%$ accuracy on CIFAR-10.

When using the multi-level refinement training scheme in combination with piecewise linear basis functions, we are able to compress nearly $50\%$ of the parameters while sacrificing only a small amount of accuracy. On a natural language modeling task, we are able to compress nearly 98% of the parameters, while still achieving good predictive accuracy. Building upon the theoretical underpinnings of numerical analysis, we demonstrate that our compression method reliably generates consistent models without requiring retraining and *without needing to revisit any data*. However, a limitation of our approach is that the implicit regularization effect introduced by the multi-level refinement training scheme can potentially reduce the accuracy of the source model.

Hence, future work should investigate improved training strategies, and explore alternative basis function sets to further improve the compression performance. We can also explore smarter strategies for compression and computational graph shortening by pursuing $hp$-adaptivity algorithms [38].

## Acknowledgments

We are grateful for the generous support from Amazon AWS and Google Cloud. NBE and MWM would like to acknowledge IARPA (contract W911NF20C0035), NSF, and ONR for providing partial support of this work. Our conclusions do not necessarily reflect the position or the policy of our sponsors, and no official endorsement should be inferred.

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
