# Stateful ODE-Nets using Basis Function Expansions
## —Supplement Materials—

## A  Proofs

### A.1  Proof of Theorem 1

In the sequel, we assert that $f$ is smooth on an interval $[0, T]$, and therefore possesses bounded derivatives of all orders on $[0, T]$, where $T > 0$. Recall that a $p$-stage explicit Runge-Kutta method for an ordinary differential equation $y'_\theta(t) = f(y_\theta(t), \theta(t))$ provides an approximation $y_{h,\theta}(t)$ of $y_\theta(t)$ for a given step size $h > 0$ through linear interpolation between the recursively generated points:

$$y_{h,\theta}(t + h) = y_{h,\theta}(t) + h\Phi_h(t, y_{h,\theta}(t)),$$

where $\Phi_h(t, y) = \sum_{i=1}^{p} b_i k_i(t, y)$ and each

$$k_i(t, y) = f\left(y + h\sum_{j=1}^{i-1} a_{ij}k_j(t, y), \theta(t + c_i h)\right).$$

It is typical to assume that $c_i = \sum_{j=1}^{i-1} a_{ij}$. For a fixed Runge-Kutta scheme $(a_{ij}, b_i, c)$, let $I_T$ denote the collection of time points where the scheme evaluates $\theta(t)$ on the interval $[0, T]$, that is,

$$I_T := [0, T] \cap \bigcup_{k \in \mathbb{Z}} \bigcup_{i=1}^{p} \{kh + c_i h\}.$$

For now, we let $\bar{\theta}(t)$ denote an arbitrary smooth function satisfying $\theta(t_j) = \bar{\theta}(t_j)$ for each $t_j \in I_T$. This ensures that the Runge-Kutta approximations $y_{h,\theta}$ and $\bar{y}_{h,\theta}$ to the ordinary differential equations

$$y'_\theta(t) = f(y_\theta(t), \theta(t))$$
$$\bar{y}'_\theta(t) = f(\bar{y}_\theta(t), \bar{\theta}(t)),$$

respectively, will coincide (that is, $y_{h,\theta} = \bar{y}_{h,\theta}$). For example, one could take $\bar{\theta}$ to be the smoothing spline satisfying

$$\bar{\theta}(t) = \arg\min_{\bar{\theta}} \left(\sum_j (\theta(t_j) - \bar{\theta}(t_j))^2 + \int_0^T \|\bar{\theta}^{(p+1)}(t)\|\mathrm{d}t\right).$$

Letting

$$\mathcal{D}_\theta^p(t) = \left\{\frac{\partial^{m+l} f_j}{\partial y^m \partial t^l}(\bar{y}_\theta(t), \bar{\theta}(t))\right\}_{j=1,\dots,d,\, k+l \le p+1},$$

Theorem II.3.2 of [18] implies the existence of a polynomial $P$ such that for any $t \in [0, T - h]$,

$$\bar{y}_\theta(t + h) = \bar{y}_\theta(t) + h\Phi_h(t, \bar{y}_\theta(t)) + h^{p+1}P \circ \mathcal{D}_\theta^p(t) + \mathcal{O}(h^{p+2}).$$

From [18, Theorem II.3.4], we have also that $y_{h,\theta}(t) = \bar{y}_\theta(t) + \mathcal{O}(h^p)$. Furthermore, by [23, §67], for any integer $m \ge 1$,

$$h^{-m}\Delta_h^m f(y, t) = \frac{\partial^m}{\partial t^m}f(y, t) + \mathcal{O}(h),$$

as $h \to 0^+$. Therefore, letting

$$\bar{\mathcal{D}}_{h,\theta}^p(t) = \left\{h^{-l}\frac{\partial^{m+l}}{\partial y^m}\Delta_h^l(y_{h,\theta}(t), \bar{\theta}(t))\right\}_{j=1,\dots,d,\, k+l \le p+1},$$

we infer that $\bar{\mathcal{D}}_{h,\theta}^p(t) - \mathcal{D}_\theta^p(t) = \mathcal{O}(h^p) + \mathcal{O}(h) = \mathcal{O}(h)$. Consequently, for any $t \in [0, T - h]$,

$$\bar{y}_\theta(t + h) = \bar{y}_\theta(t) + h\Phi_h(t, \bar{y}_\theta(t)) + h^{p+1}P \circ \bar{\mathcal{D}}_{h,\theta}^p(t) + \mathcal{O}(h^{p+2}).$$

Moving to a global estimate, [18, Theorem II.8.1] implies that for $\bar{e}_{h,\theta}(t)$ satisfying

$$\bar{e}'_{h,\theta}(t) = \frac{\partial f}{\partial y}(\bar{y}_\theta(t), \bar{\theta}(t))\bar{e}_{h,\theta}(t) + P \circ \bar{\mathcal{D}}^p_{h,\theta}(t),$$

there is, for any $t \in [0, T]$,

$$y_{h,\theta}(t) = \bar{y}_\theta(t) + h^p \bar{e}_{h,\theta}(t) + \mathcal{O}(h^{p+1}).$$

For any $t \in [0, T]$, we let $\iota_h(t) = \lfloor t/h \rfloor \cdot h$ denote the nearest point on the grid $\{0, h, 2h, \dots\}$ to $t$. Since $\bar{\theta}$ is Lipschitz continuous on $[0, T]$, $\theta(\iota_h(t)) = \bar{\theta}(\iota_h(t)) = \bar{\theta}(t) + \mathcal{O}(h)$. Therefore, by letting

$$\mathcal{D}^p_{h,\theta}(t) = \left\{ h^{-l} \frac{\partial^{m+l}}{\partial y^m} \Delta^l_h(y_{h,\theta}(t), \theta(\iota_h(t))) \right\}_{j=1,\dots,d,\ k+l \leq p+1},$$

we note that $\mathcal{D}^p_{h,\theta}(t) = \bar{\mathcal{D}}^p_{h,\theta}(t) + \mathcal{O}(h)$. Similarly, letting

$$e'_{h,\theta}(t) = \frac{\partial f}{\partial y}(\bar{y}_\theta(t), \theta(\iota_h(t)))e_{h,\theta}(t) + P \circ \mathcal{D}^p_{h,\theta}(t),$$

an application of Gronwall's inequality reveals that $\bar{e}_{h,\theta}(t) = e_{h,\theta}(t) + \mathcal{O}(h)$. Therefore,

$$y_{h,\theta}(t) = \bar{y}_\theta(t) + h^p e_{h,\theta}(t) + \mathcal{O}(h^{p+1}).$$

A Taylor expansion in $L$ finally reveals

$$L(y_{h,\theta}(t)) = L(\bar{y}_\theta(t)) + h^p \nabla L(\bar{y}_\theta(t)) \cdot e_{h,\theta}(t) + \mathcal{O}(h^{p+1}),$$

and hence the result.

## A.2 Proof of Lemma 1

Following the notation used in the proof of Theorem 1, the order conditions for a $p$-stage Runge–Kutta method ensure the existence of a function $\mathcal{E}_{h,\theta}(t)$ uniformly bounded in $h$ and depending smoothly on $\theta$, such that for any $t \in [0, T]$

$$y_{h,\theta+\epsilon\varphi}(t) = y_{\bar{\theta}+\epsilon\varphi}(t) + h^p \mathcal{E}_{h,\theta+\epsilon\varphi}(t).$$

For more details, see [18, Chapter II.3]. Therefore, as $h \to 0^+$, for any $t \in [0, T]$,

$$D_\varphi y_{h,\theta}(t) = D_\varphi \bar{y}_\theta(t) + \mathcal{O}(h^p).$$

The remainder of the proof follows by straightforward calculation. Since

$$\frac{\mathrm{d}}{\mathrm{d}\epsilon}\dot{y}_{\bar{\theta}+\epsilon\varphi}(t) = \frac{\partial f}{\partial y}(y_{\bar{\theta}+\epsilon\varphi}(t), \bar{\theta}(t) + \epsilon\varphi(t))\frac{\mathrm{d}}{\mathrm{d}\epsilon}y_{\bar{\theta}+\epsilon\varphi}(t) + \frac{\partial f}{\partial\theta}(y_{\bar{\theta}+\epsilon\varphi}(t), \bar{\theta}(t) + \epsilon\varphi(t))\varphi(t),$$

it follows that

$$\frac{\mathrm{d}}{\mathrm{d}t}D_\phi \bar{y}_\theta(t) = \frac{\partial f}{\partial y}(\bar{y}_\theta(t), \bar{\theta}(t))D_\varphi \bar{y}_\theta(t) + \frac{\partial f}{\partial\theta}(\bar{y}_\theta(t), \bar{\theta}(t))\varphi(t).$$

Since $D_\varphi \bar{y}_\theta(0) = 0$, solving this ODE reveals

$$D_\varphi \bar{y}_\theta(t) = \int_0^t e^{F_\theta(s,t)}\frac{\partial f}{\partial\theta}(\bar{y}_\theta(s), \bar{\theta}(s))\varphi(s)\mathrm{d}s,$$

where $F_\theta(s,t) = \int_s^t \frac{\partial f}{\partial y}(\bar{y}_\theta(u), \bar{\theta}(u))\mathrm{d}u$. The result now follows since $y_{h,\theta}(t) = \bar{y}_\theta(t) + \mathcal{O}(h^p)$, which in turn, implies $F_{h,\theta} = F_\theta + \mathcal{O}(h^p)$.

# B    Relationships Between Basis Functions and Prior ODE-Nets

The basis function representation of weights provides a systematic way to increase the depth-wise capacity within a single ODE-Net while using the same network unit. Using a single OdeBlock for depth is required for model transformations, such as compression, multi-level refinement, and graph shortening, to make significant changes to the model. In this section, we briefly describe how previous attempts of adding more depth to ODE-Nets with ad hoc changes to the network can be interpreted as basis functions.

For ODE-Nets, the module inside of the time integral is a neural network with time as an additional input:

$$y = \int_0^T F(x, t; \hat{\theta}) \mathrm{d}t. \tag{18}$$

Some ODE-Net implementations use $F(x, t; \hat{\theta}) = \mathcal{R}(x; \hat{\theta})$ with no explicit time dependence [13], which is as if there is a single constant basis function, $\phi(t) = 1$. Increasing the number of parameters requires stacking OdeBlocks, which is similar to adding more piecewise constant basis functions [34]. However, and importantly for us, separate OdeBlocks does not allow for integration steps to cross parameter boundaries, and does not enable compression or multi-level refinement.

In other works, e.g., [14], the time dependence is included by appending $t$ as a feature to the initial input and/or every internal layer of $\mathcal{R}$ to every other layer as well. In one perspective, this changes the structure of the recurrent unit. However, by algebraic manipulation of $t$, we can show that it is similar to a basis function representation plugged into the original recurrent unit, $\mathcal{R}(x; \theta(t, \hat{\theta}))$. Suppose the original $\mathcal{R}$ has two hidden features $x_1, x_2$, and six weights $W_{11}, W_{12}, W_{21}, W_{22}, b_1, b_2$. In matrix notation, concatenation of $t$ means that every linear transformation layer (in the $2 \times 2$ example) has two more weights and can be written as

$$y = \begin{bmatrix} W_{11} & W_{12} & W_{13} \\ W_{21} & W_{22} & W_{23} \end{bmatrix} \begin{Bmatrix} x_1 \\ x_2 \\ t \end{Bmatrix} + \begin{Bmatrix} b_1 \\ b_2 \end{Bmatrix} = \begin{bmatrix} W_{11} & W_{12} \\ W_{21} & W_{22} \end{bmatrix} \begin{Bmatrix} x_1 \\ x_2 \end{Bmatrix} + \begin{Bmatrix} b_1 + W_{13}t \\ b_2 + W_{23}t \end{Bmatrix}.$$

We can rename the third column of $W$ to be an additional basis coefficient of $b$. In tensor notation, we have effectively a representation where one of the weights is based on a simple linear function,

$$y = Wx + b(t),$$

where $W(t) = W$ is constant in time, but $b(t) = b + b_t t$. (This requires mixing basis functions for different components of $\theta$; in the main text, we only used representations where every weight used the same basis.) Thus, this unit is still similar to the original $\mathcal{R}$, but only adds another set of bias parameters as a standard ResNet unit.

As an easy extension of the above model, it is also possible to make $W$ into a linear function of $t$,

$$W(t) = W + W_t t.$$

Then, every component can be included in a parameter function $\theta(t) = \hat{\theta}_0 + \hat{\theta}_1$, where every component of the weight parameters is using the same basis set,

$$\phi_1(t) = 1,$$
$$\phi_2(t) = t.$$

These are the first two terms of a polynomial basis set which generalizes as $\phi_n(t) = t^n$. The Galerkin ODE-Nets of [34] used general polynomial terms as one basis choice.

Note that the coefficients $\hat{\theta}_1$ and $\hat{\theta}_2$ have different "units": "thetas" versus "thetas-per-second". Thus, the weight parameters for the time-coefficients need different initialization schemes, and potentially learning rates, as the weight parameters that are constant-coefficients. To the contrary, the piecewise-constant, piecewise-linear and discontinuous piecewise-linear functions we considered in the main text have basis function coefficients with the same "units," which can all use standard initialization schemes with no special consideration.

## C  Algorithm Details

### C.1  Projection

Numerical integration of the loss function equation is easy to implement, and can be exactly correct for functions with finite polynomial order with sufficient quadrature terms. We break down the domain into sub-cells and use Gaussian quadrature rules on each sub-cell. We use sub-cells because our basis sets are not smooth across control point boundaries. Specifically, we choose $N_{cell}$ as $\max(K_1, K_2)$ to line up with the finer partition, where $K$ denotes the number of basis functions. Further, we use degree 7 quadrature rules.

Given a quadrature rule with $N_{quad}$ weights $w_j$ at point $\xi_j$, we approximate the projection integral as

$$\int_0^1 f(t)\mathrm{d}t \approx \sum_{i=1}^{N_{cell}} \sum_{j=1}^{N_{quad}} w_j f(t_j) \frac{t_{i+1} - t_i}{2}, \tag{19}$$

where $t_j$ is the mapping of quadrature point $\xi_j$ from the quadrature domain $[-1, 1]$ to the cell domain $[t_i, t_{i+1}]$. This step is not performance critical, and the summations can be simplified using constant folding at compile time. The operator results in a $K_1 \times K_2$ matrix that can be applied to every parameter independently.

Now, we describe how to solve the following problem given in the main text:

$$\min_{\hat{\theta}_k^2} \int_0^T \left( \theta^1(t, \hat{\theta}^1) - \theta^2(t, \hat{\theta}^2) \right)^2 \mathrm{d}t = \min_{\hat{\theta}_k^2} \int_0^T \left( \sum_{a=1}^{K_1} \hat{\theta}_a^1 \phi_a^1(t) - \sum_{k=1}^{K_2} \hat{\theta}_k^2 \phi_k^2(t) \right)^2 \mathrm{d}t. \tag{20}$$

The projection algorithm is applied to each scalar weight of $\theta$ independently. Let $X_a$ be one scalar component of $\hat{\theta}_a^1$ ($a = 1, \ldots, K_1$), and $Y_k$ be the matching scalar component of $\hat{\theta}_k^2$ ($k = 1, \ldots, K_2$). The loss function is defined on the basis coefficients of each scalar weight as

$$L(X, Y) = \sum_{i=1}^{N_{cell}} \sum_{j=1}^{N_{quad}} \frac{w_j(t_{i+1} - t_i)}{2} \left( \sum_{a=1}^{K_1} X_a \phi_a^1(t) - \sum_{k=1}^{K_2} Y_k \phi_k^2(t) \right)^2. \tag{21}$$

This function is quadratic and can be solved with one linear solve, $Y = -H^{-1}G(x)$ with

$$H_{jk} = \frac{\partial^2 L}{\partial \hat{\theta}^2} = \texttt{Integrate}\left( \phi_j^2(t)\phi_k^2(t) \right) \tag{22}$$

$$G_j(X) = \frac{\partial L}{\partial \hat{\theta}} = \texttt{Integrate}\left( \phi_j^2(t) \sum_{a=1}^{K_1} \phi_a^1(t) X_a \right), \tag{23}$$

where $H$ is a $K_2 \times K_2$ Hessian matrix, and $G(X)$ is a $K_2$ gradient vector. (In practice, we merely implement $L(X, Y)$ in Python, using NumPy to obtain the quadrature weights, and use JAX's automatic differentiation to evaluate the Hessian $H$ and gradient $G$.) The set of basis function coefficients $\hat{\theta}^1$ has $K_1 \times N_{\mathcal{R}}$ components, and $\hat{\theta}^2$ has $K_2 \times N_{\mathcal{R}}$ components. This linear solve is applied to every tensor component of $\hat{\theta}$, in a loop needing $N_{\mathcal{R}}$ iterations where $X$ is a different component of $\hat{\theta}^1$. Note that $H$ is independent of $X$, and $G(X)$ is linear in $X$ and can be written as $G(X) = RX$, where $R$ is a $K_2 \times K_1$ matrix. The loop can be efficiently carried out by pre-factorizing $H$ once, then applying the back-substitution to each column of the the matrix $R$ to obtain a matrix $A = H^{-1}R$. Then, $A$ can be applied to every component column in $\hat{\theta}^1$, obtaining a linear operation

$$\underbrace{[\hat{\theta}^2]}_{K_2 \times N_{\mathcal{R}}} = -[[\underbrace{[H^{-1}]}_{K_2 \times K_2} \underbrace{[G]}_{K_2 \times K_1}] \underbrace{[\hat{\theta}^1]}_{K_1 \times N_{\mathcal{R}}}. \tag{24}$$

Projection of the updated state point cloud onto the basis in Equation (13) can be solved with the same algorithm by letting $L(X, Y)$ equal to the minimization objection. Gaussian quadrature is not needed to calculate $L$, and automatic differentiation can be directly applied to the least-squares summation over the point cloud.

**A Note on Complexity.** Basis transformations are data agnostic, i.e., they operate directly on the parameter coefficients. The total number of parameters for the two basis function models is $N_{\mathcal{R}}K_1$ and $N_{\mathcal{R}}K_2$, where $N_{\mathcal{R}}$ is the number of coordinates of $\mathcal{R}$. Projection requires one matrix factorization and $N_{\mathcal{R}}$ applications of the factorization; the general-case complexity is $\mathcal{O}\left((K_2)^3 + (K_2)^2K_1 + N_{\mathcal{R}}K_1K_2\right)$. Interpolation requires $K_2$ evaluations of $\phi^1$ for each coordinate; the general-case complexity of interpolation is $\mathcal{O}\left(N_{\mathcal{R}}K_2K_1\right)$. $K_1$ and $K_2$ are proportional to the number of layers in a model and thus relatively small numbers, as compared to $N_{\mathcal{R}}$.

## C.2 Stateful Normalization Layer

Algorithm 1 lists the forward pass during training. Tracking a list of updated module states (i.e. BatchNorm statistics) at times $t_i$ is fused with integrating the ODE-Net forwards in time. By using a fixed times integration scheme (with constant $\Delta t$), this algorithm yields a static computational graph. In practice, by using basis functions with compact support, the computational graph can further optimized by interleaving the `Project` calculation with the integration loop. At inference time, the state parameters are fixed, so it is not necessary to compute $\mathcal{R}_s$, save the list `States`, or perform projection to $\hat{\theta}^{s*}$.

---

**Algorithm 1:** `StatefulOdeBlock` accumulates and projects state updates from the Runge-Kutta forward pass.

---

**Data:** Gradient and state parameters $\hat{\theta}^g$, $\hat{\theta}^s$, Input $x_{in}$.
Initialize `States` $= \{\}$;
Let $x = x_{in}$;
**for** $t = 0$, $t < T$, $t = t + \Delta t$ **do**
    **foreach** *Runge-Kutta Stage i* **do**
        Let $t_i = t + c_i\Delta t$;
        $x_i = \sum_j x + \Delta t a_{ij}k_j$;
        $k_i = \mathcal{R}_x(\theta^g(t_i,\hat{\theta}^g), \theta^s(t_i,\hat{\theta}^s), x_i)$;
        $\bar{\theta}_i^s = \mathcal{R}_s(\theta^g(t_i,\hat{\theta}^g), \theta^s(t_i,\hat{\theta}^s), x_i)$;
        `States.append(`$\{t_i, \bar{\theta}_s^i\}$`)`;
    **end**
    Let $x = x + \Delta t \sum_i b_i k_i$;
**end**
$x_{out} = x$;
$\hat{\theta}^{s*} = $ `Project(States`$, \phi, K)$;
**Forward pass outputs:** $x_{out}$, $\hat{\theta}^{s*}$;
**Use** $x_{out}$ to compute $loss$.;
**Backward pass:** Trace $\partial x_{out}/\partial\hat{\theta}^g$;
**Update gradient parameters:** $\theta^g \leftarrow OptimizerStep(\hat{\theta}^g, \partial loss/\partial\hat{\theta}^g)$;
**Update state parameters:** $\hat{\theta}^s \leftarrow \hat{\theta}^{s*}$;

---

## C.3 Refinement Training

Piecewise constant basis functions yield a simple scheme to make a neural network deeper and increase the number of parameters: double the number of basis functions, and copy the weights to new grid. This insight was used by [6] and [37] to accelerate training. In these prior works, network refinement was implemented by copying and re-scaling discrete network objects, or expanding tensor dimensions.

Instead, we view this problem through the lens of basis function interpolation and projection. The procedure can be thought of as projecting or interpolating to a basis set with more functions:

$$\hat{\theta}^{refined} = \texttt{Interpolate}\left(\left(\sum_{k=1}^{K}\hat{\theta}_k\phi_k(t)\right), \texttt{next}(\phi), \texttt{next}(K)\right), \tag{25}$$

where `next(·)` is an arbitrary schedule picking the new basis functions. `Project` can also be used instead of `Interpolate`. The multi-level refinement training of [6, 37] is equivalent to interpolating a piecewise constant basis set to twice as many basis functions by evaluating $\hat{\theta}_k^2 = \theta^1(t_k)$ at new cell centers $t_k = T(k-1)/(K_2-1)$. For exactly doubling the number of parameters, the evaluation of $\sum \hat{\theta}^1 \phi^1(t_k)$ can be simplified as a vector of twice as many basis coefficients,

$$\hat{\theta}^2 = \{\hat{\theta}_1^1, \hat{\theta}_1^1, \hat{\theta}_2^1, \hat{\theta}_2^1 ... \hat{\theta}_{K_1}^1, \hat{\theta}_{K_1}^1\}, \ K_2 = 2K_1. \tag{26}$$

The "splitting" concept can be extended to piecewise linear basis functions by adding an additional point into the midpoint of cells. The midpoint control point evaluates to the average of the parameters at the endpoints. This results in the new list of coefficients

$$\hat{\theta}^2 = \left\{ \hat{\theta}_1^1, \frac{\hat{\theta}_1^1 + \hat{\theta}_2^1}{2}, \hat{\theta}_2^1, \frac{\hat{\theta}_2^1 + \hat{\theta}_3^1}{2}, \hat{\theta}_3^1 ... \hat{\theta}_{K_1-1}^1, \frac{\hat{\theta}_{K_1-1}^1 + \hat{\theta}_{K_1}^1}{2}, \hat{\theta}_{K_1}^1 \right\}, \ K_2 = 2K_1 - 1. \tag{27}$$

Both of these splitting-based interpolation schemes are exactly correct for piecewise constant and piecewise linear basis functions. Multi-level refinement training can be applied to any basis functions and any pattern for increasing $K$ by using the general interpolation and projection methods.

To make sure that the ODE integration in the forward pass visits all of the new parameters, we also increase the number of steps $N_T$ after refinement. A sketch of a training regiment using interpolation is shown in Algorithm 2.

---

**Algorithm 2:** Sketch of a training algorithm using multi-level refinement training as an interpolation step. The interpolation step is applied to all of the *StatefulOdeBlock*s inside of the model. The same interpolation is applied to gradient parameters and state parameters. Interpolation can be replaced by projection.

---

**def** $\texttt{next}(K)$ = Pattern to increase basis functions; e.g. $\texttt{next}(K) = 2K$.;
**def** $loss(model, \hat{\theta}^g, \hat{\theta}^s, batch) = logits(Y, model(\hat{\theta}^g, \hat{\theta}^s, X))$;
$model = \texttt{ContinuousModel}(N_{T,initial}, \texttt{scheme}, K_{initial}, \phi)$ ;
$\hat{\theta}^g, \hat{\theta}^s = \texttt{Initialize}(model)$ ;
$\texttt{Optimizer} = \texttt{MakeOptimizer}(loss, \hat{\theta}^g)$;
**for** $e \in [1, ...N_{epochs}]$ **do**
   **if** $e \in$ *Refinement Epochs* **then**
      $\hat{\theta}^g = \texttt{Interpolate}(\hat{\theta}^g, \phi, \texttt{next}(model.K)$;
      $\hat{\theta}^s = \texttt{Interpolate}(\hat{\theta}^s, \phi, \texttt{next}(model.K))$;
      $model.K = \texttt{next}(model.K)$    # Update the model hyperparmeters to track $K$.;
      $model.N_T = \texttt{next}(model.N_t)$    # Also increase the number of integration steps.;
      $\texttt{Optimizer} = \texttt{MakeOptimizer}(loss, \hat{\theta}^g)$;
   **for** $batch \in$ *Training Data* **do**
      $l, \hat{\theta}^s = loss(model, \hat{\theta}^g, \hat{\theta}^s, batch)$;
      $\hat{\theta}^g = \texttt{Optimizer}(\hat{\theta}^g, \nabla l)$ ;
   Save Checkpoint$(\hat{\theta}^g, \hat{\theta}^s)$;

---

# D   Additional Results

## D.1   Results for CIFAR-100.

Table 5 tabulates results for applying the same methodology to CIFAR-100. Again, we consider two configurations: (c1) is a model trained with refinement training, which has piecewise linear basis functions; (c2) is a model trained without refinement training, which has piecewise constant basis functions. Similar to the results for CIFAR-10, we see that model (c2) achieves high predictive accuracy, however, the compression performance is poor. Our model (c1) is less accurate, but we are able to compress the number of parameters by about 41% with less than 1% loss of accuracy on average.

Table 5: Compression performance and test accuracy of Deep ODE-Nets on CIFAR-100.

| Model | Best | Average | Min | # Parameters | Compression |
|---|---|---|---|---|---|
| Wide-ResNet [46] | - | 78.8% | - | 17.2M | - |
| ResNet-122-i [6] | - | 73.2% | - | 7.7M | - |
| Wide-ContinuousNet [37] | 79.7% | 78.8% | 78.2% | 13.6M | - |
| Stateful ODE-Net (c1) | 76.2% | 76.9% | 75.5% | 15.2M | - |
| $\hookrightarrow$ (compressed) | 75.9% | 75.6% | 75.2% | 9.0M | 41% |
| Stateful ODE-Net (c2) | **79.9%** | **79.1%** | **78.5%** | 13.6M | - |
| $\hookrightarrow$ (compressed) | 52.7% | 48.9% | 39.5% | 9.0M | 34% |

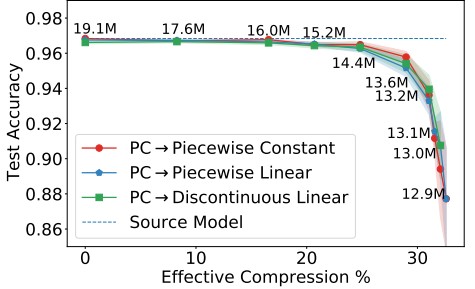

(a) Compressing basis coefficients with fixed $N_T$.

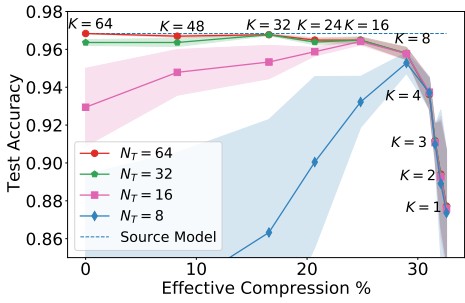

(b) Decreasing $N_T$ using the red line in (a).

Figure 6: In (a), we show the compression of a continuous-in-depth transformer for part-of-speech tagging. Discounting the 12.8M parameters in the embedding table and only considering the parameters in the transformer layers and classifier, the smallest model ($K = 1$) achieves 98.3% compression compared to the source model ($K = 64$). In (b) it is observed that the model graph can be shortened.

## D.2 Continuous Transformers Applied to German-HDT

We applied the same architecture as Section 6.3 to the German-HDT dataset as well [2]. This dataset has a vocabulary size of 100k (vs. 19.5k) and 57 labels (vs. 53). All other hyperparameters and learning rate schedule are the same as the original transformer. The final trained source model also has $K = 64$ basis functions. The compression and graph shortening experiments are repeated for this cohort of models, again sampled from 8 seeds, and the results are shownin Figure 6. We also perform the procedure of projecting the source model with $K = 64$ down to the smallest possible model, $K = 1$. Because of the increased vocabulary size, the embedding table is much larger, and thus the effective compression at $K = 1$ is only 32%. We observe the same behavior as in the English-GUM dataset. The resulting models are more accurate for this dataset, but the relative performance trade-off with respect to compression is similar.

# E    Model Configurations

Here, we present details about the model architectures that we used for our experiments. (The provided software implementation provides further details.)

## E.1    Image Classification Networks

At a high-level, our models are composed of two types of blocks that allow us to construct computational graphs that are similar to those of ResNets.

- The *StatefulOdeBlock* can be regarded as a drop-in replacement for standard ResNet blocks. The ODE rate, $\mathcal{R}$ takes the form of the residual units. For instance, for image classification tasks, this block consists of two convolutional layers in combination with pre-activations and BatchNorms. Specifically, we use the following structure:

$$\mathcal{R} = x \to BN \to ReLU \to Conv \to BN \to ReLU \to Conv.$$

However, the user can define any other structure that is suitable for a particular task at hand.

- The *StichBlock* has a similar form as compared to the OdeBlock (two convolutional layers in combination with pre-activations), but in addition this block allows us to perform operations such as down-sampling by replacing the skip connection with a stride of 2.

The number of channels and strides can be chosen the same way as for discrete ResNet configurations. In the following we explains the detailed structure of the different models used in our experiments. For simplicity, we omit batch normalization layers and non-liner activations.

**Shallow ODE-Net for MNIST.** Table 6 describes the initial configuration for our MNIST experiments. Here the initial architecture consists of a convolutional layer followed by a *StichBlock* and a *StatefulOdeBlock* with 12 channels. During training we increase the number of basis functions in the *StatefulOdeBlock* from 1 to 8, using the multi-level refinement training scheme.

*Training details.* We train this model for 90 epochs with initial learning rate 0.1 and RK4. We refine the model at epochs 20, 50, and 80. We use batch size 128 and stochastic gradient descent with momentum 0.9 and weight decay of $0.0005$ for training.

Table 6: Summary of architecture used for MNIST.

| Name | output size | Channel In / Out | Kernel Size | Stride | Residual |
|---|---|---|---|---|---|
| conv1 | 28×28 | 1 / 12 | 3×3 | 1 | No |
| StichBlock_1 | 28×28 | 12 / 12 | $\begin{bmatrix} 3\times3 \\ 3\times3 \end{bmatrix}$ | 1 | Yes |
| StatefulOdeBlock_1 | 28×28 | 12 / 12 | $\begin{bmatrix} 3\times3 \\ 3\times3 \end{bmatrix}$ | - | Yes |

| Name | Kernel Size | Stride |
|---|---|---|
| average pool | 8×8 | 8 |

| Name | input size | output size |
|---|---|---|
| FC | - | 10 |

**Shallow ODE-Net for CIFAR-10.** Table 7 describes the initial configuration for our CIFAR-10 experiments. Here the initial architecture consists of a convolutional layer followed by a *StichBlock* and a *StatefulOdeBlock* with 16 channels, followed by another *StichBlock* and *StatefulOdeBlock* with 32 channels. The second *StichBlock* has stride 2 and performs a down-sampling operation. During training we increase the number of basis functions in both *StatefulOdeBlock*s from 1 to 8, using the multi-level refinement training scheme. That is, each *StatefulOdeBlock* has a separate basis function set, but both have the same $K$.

*Training details.* We train this model for 200 epochs with initial learning rate 0.1 and RK4. We refine the model at epochs 50, 110, and 150. We use a batch size of 128 and stochastic gradient descent with momentum 0.9 and weight decay of $0.0005$ for training.

Table 7: Summary of shallow architecture used for CIFAR-10.

| Name | output size | Channel In / Out | Kernel Size | Stride | Residual |
|---|---|---|---|---|---|
| conv1 | 28×28 | 3 / 16 | 3×3 | 1 | No |
| StichBlock_1 | 32×32 | 16 / 16 | $\begin{bmatrix} 3\times3 \\ 3\times3 \end{bmatrix}$ | 1 | Yes |
| StatefulOdeBlock_1 | 32×32 | 16 / 16 | $\begin{bmatrix} 3\times3 \\ 3\times3 \end{bmatrix}$ | - | Yes |
| StichBlock_2 | 16×16 | 16 / 32 | $\begin{bmatrix} 3\times3 \\ 3\times3 \end{bmatrix}$ | 2 | Yes |
| StatefulOdeBlock_2 | 16×16 | 32 / 32 | $\begin{bmatrix} 3\times3 \\ 3\times3 \end{bmatrix}$ | - | Yes |

| Name | Kernel Size | Stride |
|---|---|---|
| average pool | 8×8 | 8 |

| Name | input size | output size |
|---|---|---|
| FC | - | 10 |

**Deep ODE-Net for CIFAR-10.** Table 7 describes the initial configuration for our CIFAR-10 experiments using deep ODE-Nets. Here the initial architecture consists of a convolutional layer

followed by a *StichBlock* and a *StatefulOdeBlock* with 16 channels, followed by another *StichBlock* and *StatefulOdeBlock* with 32 channels, followed by another *StichBlock* and *StatefulOdeBlock* with 64 channels. The second and third *StichBlock* have stride 2 and perform down-sampling operations. During training we increase the number of basis functions in each of the three *StatefulOdeBlock*s from 1 to 16, using the multi-level refinement training scheme. The three *StatefulOdeBlock* uniformly have the same number of basis functions.

*Training details.* We train this model for 200 epochs with initial learning rate 0.1 and RK4. We refine the model at epochs 20, 40, 70, and 90. We use batch size 128 and stochastic gradient descent with momentum 0.9 and weight decay of $0.0005$ for training.

**Deep ODE-Net for CIFAR-100.** For our CIFAR-100 experiments we use the same initial configuration as for CIFAR-10 in Table 7, with the difference that we increase the number of channels by a factor of $4$. Then, during training we increase the number of basis functions in each of the *StatefulOdeBlock*s from 1 to 8, using the multi-level refinement training scheme.

*Training details.* We train this model for 200 epochs with initial learning rate 0.1 and RK4. We refine the model at epochs 40, 70, and 90. We use batch size 128 and stochastic gradient descent with momentum 0.9 for training.

Table 8: Summary of deep shallow architecture used for CIFAR-10.

| Name | output size | Channel In / Out | Kernel Size | Stride | Residual |
|---|---|---|---|---|---|
| conv1 | 28×28 | 3 / 16 | 3×3 | 1 | No |
| StichBlock_1 | 32×32 | 16 / 16 | $\begin{bmatrix} 3\times3 \\ 3\times3 \end{bmatrix}$ | 1 | Yes |
| StatefulOdeBlock_1 | 32×32 | 16 / 16 | $\begin{bmatrix} 3\times3 \\ 3\times3 \end{bmatrix}$ | - | Yes |
| StichBlock_2 | 16×16 | 16 / 32 | $\begin{bmatrix} 3\times3 \\ 3\times3 \end{bmatrix}$ | 2 | Yes |
| StatefulOdeBlock_2 | 16×16 | 32 / 32 | $\begin{bmatrix} 3\times3 \\ 3\times3 \end{bmatrix}$ | - | Yes |
| StichBlock_3 | 8×8 | 32 / 64 | $\begin{bmatrix} 3\times3 \\ 3\times3 \end{bmatrix}$ | 2 | Yes |
| StatefulOdeBlock_3 | 8×8 | 64 / 64 | $\begin{bmatrix} 3\times3 \\ 3\times3 \end{bmatrix}$ | - | Yes |

| Name | Kernel Size | Stride |
|---|---|---|
| average pool | 8×8 | 8 |

| Name | input size | output size |
|---|---|---|
| FC | - | 10 |

## E.2 Part-of-Speech Tagging Networks

For our part-of-speech tagging experiments, we use the continuous transformer illustrated in Figure 1 of the main text. This network only has four major components: the input sequence is fed into an embedding table loop up, then concatenated with a position embedding, and then fed into a single *StatefulOdeBlock*. The output embeddings are used by a fully connected layer to classify each individual token in the sequence. The network structure of the module $\mathcal{R}$ for the encoder is diagrammed on the right side of Figure 1. Note that the $\mathrm{d}x/\mathrm{d}t$ encoder has slightly rearranged skip connections from the discrete encoder. The Self Attention and MLP blocks apply layer normalization to their inputs. In our implementation, we did not use additional dropout layers.

Every vector in the embedding table is of size 128. The kernel dimensions of the query, key, and value kernels of the self attention are 128. The MLP is a shallow network with 128 dimensions.

*Training details.* We train this model for 35000 iterations, with a batch size of 64. We use Adam for training. The learning rate follows an inverse square-root schedule, with an initial value of 0.1 and a linear ramp up over the first 8000 steps. The optimizer parameters are $\beta_1 = 0.9$, $\beta_2 = 0.98$, and $\epsilon = 10^{-9}$, with weight decay of 0.1. The refinement method is applied at steps 1000, 2000, 3000, 4000, 5000, and 6000 to grow the basis set from $K = 1$ to $K = 64$.