# OpenReview forum: "Stateful ODE-Nets using Basis Function Expansions"
_NeurIPS.cc/2021/Conference — NeurIPS 2021 Poster_

### Official Review · Reviewer_KZk9 · 2021-07-16

**Rating:** 6
**Confidence:** 3

**Summary:**

The paper introduces a new neural ODE with time-continuos depth and time-dependent parameterization of the network. The time-dependent parameterization is composed of a linear combination of time-dependent basis function and learnable “base parameters”.

By relying on this continuous reparameterization using basis function the authors highlight the close connection with the compressibility of the network by doing a basis change during inference time to a lower-dimensional space of basis function, where the “base parameters” in the lower dimensional space can be computed either via projection or linear interpolation from the higher-dimensional basis space.

Additionally, the authors introduces a novel notion of time-continuous batch normalization and they present theoretical results pertaining to the implicit regularization potential of their model when solved via RK-methods and empirical results for image classification and time-continuous analogues of transformers.

**Ethical Concerns:**

No concerns.

**Limitations And Societal Impact:**

See above weaknesses especially with regards to the refinement in the training process.

Additionally, I glanced over Section 4.1 in the appendix regarding the implementational details of the projection step. It seems to require various derivatives including a Hessian computation. How does the projection step scale with the model? Is that something that’s feasible for larger models and datasets? What is the practical runtime for this step? It would be good to understand that.

**Main Review:**

## Score

Overall, I definitely appreciated the novelty and the rigor of the proposed time-continuous model and I believe there is scientific merit and the potential to influence the broader community. However, I am little concerned about the current presentation of the results. Specifically, they are covering lots of grounds from introducing a new type of model to theoretical results to compression to empirical results. Consequently, some of the presentation is very superficial and it’s hard to understand the important details that might help other researcher pick up on this work and build upon it.

So in short, I imagine the core contribution to be quite interesting but this has to be backed by the corresponding depth and detailed discussions of the method in the paper. Right now, there is lots of breadth and little depth across the sections.


## Ways to Improve My Score

Please address the weaknesses I am outlining below. I would specifically focus on trying to clean up some of the writing and re-structuring it to be able to go into depth about individual aspects of the work while potentially removing other aspects.

## Strengths

* Intuitive and powerful model that promises a lot of flexibility down the road.

* Interesting experimental results pertaining both to image classification and NLP/transformers.

* Good introduction and explanation about the concept of basis functions.

* I enjoyed learning about the combination of using a NN parameterization based on basis functions  + time-continuous models.

## Weaknesses

* By far the biggest weakness to me is that there is so much going on that it is very hard to formulate a take-away message at the end of the paper. Sure, I might be able to download the code, re-run the experiments, and modify the models a little bit to adapt them to my task. But I don’t think the papers contributes a deep understanding of the model that someone could leverage in their own research in order to come up with new and innovative solutions.

* Introduction: I don’t think it’s really useful to touch upon so many concepts in the introduction. Try re-structuring the core concepts, simplifying them, shortening the intro, and then go into more details about each aspect later on in the paper.

* The time-continuous batch normalization: This concept has been mentioned many times throughout the sections and I suppose I can somewhat imagine how it works but overall the language of the paper never goes beyond the conceptual idea of what the authors want to achieve. For example, looking at Algorithm 1 I cannot figure out where $\hat\theta^{s*}$ is being used? It seems to be an output of the algorithm but what are we doing with it? Is it part of the model as a batch norm layer? Where would this layer be? It’s confusing but still seems to be very important so it would be crucial to understand how this fits into the model.

* I cannot really find a take-away message from Section 5. It seems to be a somewhat interesting result from the space of dynamical systems and uses a lot of very atypical concepts that I cannot imagine are common knowledge among the broader NeurIPS community (or even the more narrow neural ODE community). If you really want to include these theoretical results, please simplify the section in the main paper and put a much more in-depth discussion into the appendix. In the current shape, I am left to simply take the authors’ word for granted about the connection between their theorem and the implicit regularization potential without really being able to back it up myself.

* The experiments are good but I would ask the authors to significantly expand upon the concept of (multi-level) refinement. This seems to be a central empirical “trick” in order to make their model work in practice. What exactly is the concept behind it? How to tune the different levels? What is the crucial insight there that makes the difference in the empirical performance?

* Choice of basis functions: The basis functions seem overly simplistic (either convex combination or piecewise). What about other basis functions and what constitutes the motivation for these basis functions? The concept and the notation are very heavy in order to introduce these basis functions. So is there any benefit beyond these simple basis functions?

* Related to my question above: It appears to me that the authors rely on a fixed-step solver like RK4. In addition, they often use piecewise-constant basis functions. My question is how this model even differs from a regular NN that explicitly models multiple layers? Specifically, if for each time step in RK4 there is a different “piecewise-constant basis function” that is activated this implies that at any given depth you call the _same layer architecture_ with a _distinct set of parameters_ much like you would simply repeat the same layer architecture with separately trained parameters in a regular NN with residual connections? And even if there is more time steps in RK4 than there are basis functions it very much still seems to boil down to a regular NN with a little bit of parameter sharing, right? And same goes for the piecewise linear basis functions, right? I am not saying that the proposed formulation isn’t more expressive in theory but it seems that all instantiations of the model framework are essentially very close to  a regular network with multiple layers of the same layer architecture stacked on top of each other. Could the authors comment on this observation?

* I would also appreciate if the authors could spend more time in the introduction and the related work on how their work differs from prior work. It seems basis functions have been explored before from what I understand from their paper and also of course neural ODEs are not a new concept. So it would be very helpful to contextualize their work within existing work more clearly and delineate their contributions.

---

## Update after rebuttal

I am raising my score to 6 notwithstanding some of the open issues that I feel need to be addressed, whether during a potential camera-ready version of the paper or a potential re-submission depending on the outcome of the review process. More details can be found below in my discussion with the authors.

**Time Spent Reviewing:**

6

---

> ### Author Response · Authors · 2021-08-11
> **Response to Reviewer KZk9 (Part 1)**
>
> We thank the reviewer for the very detailed response and excellent suggestions. We greatly appreciate the assessment that our work presents novel ideas and that the reviewer believes that **there is scientific merit** and our methods have **the potential to influence the broader community**. The suggestions of the reviewer greatly helped us to revise and restructure the presentation of work (e.g., we have streamlined our introduction, extended the related work section and expanded the discussion of the continuous-time batch normalization layer and better highlighted the importance of Theorem 1 for our work).
>
>
> *Q: By far the biggest weakness to me is that there is so much going on that it is very hard to formulate a take-away message at the end of the paper. Sure, I might be able to download the code, re-run the experiments, and modify the models a little bit to adapt them to my task. But I don’t think the papers contributes a deep understanding of the model that someone could leverage in their own research in order to come up with new and innovative solutions.*
>
> A: We agree that we present a range of novel innovations in this paper while being constrained by the tight page limit. We can extend the depth of discussion of the innovations that we present.
>
> Further, the supplementary code implementation included provides more than just the ability to re-run the experiments. The library contains a plug-and-play StatefulOdeBlock module that can be added into any model using Jax (not just Neural ODEs). We are confident that the library can be used by Neural ODE researchers to get SOTA performance on other problems, which would help Neural ODEs find practical applications.
>
> *Q: Introduction: I don’t think it’s really useful to touch upon so many concepts in the introduction. Try re-structuring the core concepts, simplifying them, shortening the intro, and then go into more details about each aspect later on in the paper.*
>
> A: We have streamlined the introduction and provide a better high-level overview of our work and contributions. However, we think it is important to present the formulation of our novel stateful neural ODE model in the introduction, which is in line with the style of NeurIPS.
>
> *Q: The time-continuous batch normalization: This concept has been mentioned many times throughout the sections and I suppose I can somewhat imagine how it works but overall the language of the paper never goes beyond the conceptual idea of what the authors want to achieve. For example, looking at Algorithm 1 I cannot figure out where ^s! is being used? It seems to be an output of the algorithm but what are we doing with it? Is it part of the model as a batch norm layer? Where would this layer be? It’s confusing but still seems to be very important so it would be crucial to understand how this fits into the model.*
>
> A: We agree that the underlying mechanics are non-trivial. Hence, with the reviewers excellent suggestions, we have extended the discussion on time-continuous batch normalization in the revised version. We strongly believe that this in depth-discussion in combination with the presented Algorithm and the provided code in Jax, will enable the community to use and adapt our ideas in a range of applications.
>
> As to the specific question, thank you for the suggestions for clarification. The notation was difficult, and we have added an additional description of how state updates work in training.
> $\hat{\theta}^{s}$ are the basis function coefficients to the continuous state params, $\theta^s(t)$. During a training step, these are updated by the BatchNorm and projected back onto the basis set by Algorithm 1.
> $\hat{\theta}^{s*}$ are the “updated” values after the forward pass during a training step, which are assigned back to $\hat{\theta}^{s}$.
> We have added the following two equations to the end of Algorithm 1, which shows how $\hat{\theta}^{s*}$ is used to close the training loop:
> New gradient-parameter basis coefficients:
>   $\hat{\theta}^g = OptimizerStep( \partial loss / \partial \hat{\theta}^g )$
> New state-parameter basis coefficients:
>   $\hat{\theta}^s = \hat{\theta}^{s*}$
>
>
> *Q: I cannot really find a take-away message from Section 5. It seems to be a somewhat interesting result from the space of dynamical systems and uses a lot of very atypical concepts that I cannot imagine are common knowledge among the broader NeurIPS community (or even the more narrow neural ODE community). If you really want to include these theoretical results, please simplify the section in the main paper and put a much more in-depth discussion into the appendix. In the current shape, I am left to simply take the authors’ word for granted about the connection between their theorem and the implicit regularization potential without really being able to back it up myself.*
>
> A: We are grateful for the feedback and appreciate the assessment that the presented statements are interesting and nontrivial. We are happy to include further details surrounding the main theorem and its interpretations.
>
> The result draws from the error expansions for discretizations of ODEs. The central takeaway is that the precise form of the error in the discretization may benefit compression --- more so for some solvers than others --- because it implicitly regularizes against the derivatives with respect to the weights. In this regard, our Theorem 1 is not so atypical in the implicit regularization literature. We do recognize that our discussion after the theorem, particularly with sensitivity as a Gateaux derivative, is more involved and can be fleshed out in Supplementary Material, summarized in the following way:
>
> “In the Supplementary Material, we show that the capacity for compression in $\theta$ decreases monotonically with decreasing derivatives of the integrand $f$ in each of its arguments. These are precisely the objects that appear in the error term $e_{h,\theta}$ in Theorem 1. Therefore, better compression can be achieved in one of two ways.
>
> * (I) A higher-order integrator is used, causing higher-order derivatives to appear in the implicit regularization term $e_{h,\theta}$. The more derivatives, the stronger the effect.
>
> * (II) A larger step size $h$ is used. Since doing so can lead to stability concerns during training, we later consider a refinement training scheme [5,47] where $h$ is slowly reduced.
>
> In Figure 3, we verify strategy (I), showing that the 4th-order RK4 scheme exhibits improved test accuracy for higher compression ratios over the 1st-order Euler scheme. Unfortunately, higher-order integrators typically increase the runtime on the order of $\mathcal{O}(p)$. Therefore, some of our later experiments will focus on strategy (II), which avoids this issue.”
>
> *Q: The experiments are good but I would ask the authors to significantly expand upon the concept of (multi-level) refinement. This seems to be a central empirical “trick” in order to make their model work in practice. What exactly is the concept behind it? How to tune the different levels? What is the crucial insight there that makes the difference in the empirical performance?*
>
> A: We agree with the reviewer’s understanding that multi-level refinement is an important factor. We did not include a detailed discussion and analysis of the algorithm because it has been previously proposed by [1] and [2], and we thought that these results are well known within the community. However, we do think that we propose a generalized algorithm to the established multi-level refinement algorithm by extending it to other basis functions and growth factors that are not powers of two.
>
> The core concept of the multi-level refinement algorithm is just projection/interpolation to more coefficients, i.e., during training we start with a small model that we slowly grow. The user has to define the epochs at which the model is projected onto more coefficients. The tuning parameter is a schedule of epochs at which to apply refinement (similar to a learning rate schedule.)
>
> There are two important aspects of this training scheme. (a) The process introduces implicit regularization by smoothing the function on the scale of $\Delta T$ ($h$ in the theorem); note, that a consequence of Theorem 1 is that implicit regularization is required in order to be able to compress the model. The multi-level refinement scheme is thus beneficial for obtaining a highly-compressible model. (b) The scheme can be seen as the reverse of the compression step (i.e., compression is projection/interpolation to fewer coefficients).
>
> To make our work better self-contained and to address the reviewer's concern, we will provide more details about the multi-level refinement scheme in the appendix.

---

> > ### Author Response · Authors · 2021-08-11
> > **Part 2**
> >
> > *Q: Choice of basis functions: The basis functions seem overly simplistic (either convex combination or piecewise). What about other basis functions and what constitutes the motivation for these basis functions? The concept and the notation are very heavy in order to introduce these basis functions. So is there any benefit beyond these simple basis functions?*
> >
> > A: We intentionally chose three simple bases that lead to models that achieve state-of-the-art performance and to simplify the discussion. We added the following context and reasoning to Section 3:
> > * These functions correspond to numerical algorithms that many readers would be familiar with:
> >    * Piecewise constant: ResNets, Finite Volume Methods
> >    * Piecewise linear: 1st order Finite Element Methods (Galerkin methods)
> >    * Discontinuous piecewise linear: 1st order Discontinuous Galerkin Methods
> >
> > * Finite Element / Finite Volume / Discontinuous Galerkin methods were the inspiration for the previously established multi-level refinement algorithm [1,2].
> >
> > * The functions have compact support, so that one time-step only uses one or two coefficients. The computational cost is thus linear with depth, O(N_t * (1 or 2)), for any N_basis. In contrast, spectral basis functions every time-step need to access all coefficients of the basis function. Basis functions without compact support have the following disadvantages:
> >     * More FLOPS are required to compute \theta(t): the cost of the forward pass is quadratic with depth! O(N_t * N_basis)
> >     * Poor cache performance: every step accesses every parameter.
> >     * For many spectral functions, such as those considered by [3], the condition number grows with the number of basis.
> >
> > *Q: Related to my question above: It appears to me that the authors rely on a fixed-step solver like RK4. In addition, they often use piecewise-constant basis functions. My question is how this model even differs from a regular NN that explicitly models multiple layers? Specifically, if for each time step in RK4 there is a different “piecewise-constant basis function” that is activated this implies that at any given depth you call the same layer architecture with a distinct set of parameters much like you would simply repeat the same layer architecture with separately trained parameters in a regular NN with residual connections? And even if there is more time steps in RK4 than there are basis functions it very much still seems to boil down to a regular NN with a little bit of parameter sharing, right? And same goes for the piecewise linear basis functions, right? I am not saying that the proposed formulation isn’t more expressive in theory but it seems that all instantiations of the model framework are essentially very close to a regular network with multiple layers of the same layer architecture stacked on top of each other. Could the authors comment on this observation?*
> >
> > A: This is an excellent observation and indeed very true. Our paper points that out several times (e.g., line 162 and 251). But we are happy to better highlight this observation in the revised version, if the reviewer thinks it is important.
> >
> > These are just observations about the algebraic structure of the model graphs. However, the algebraic structure is only half of the story: the parameters are also important. A primary contribution of our work is theoretical and empirical analysis to design a training regiment to learn the right parameters that also satisfy ODE-qualities to take advantage of our proposed techniques.
> >
> >
> > *Q: I would also appreciate if the authors could spend more time in the introduction and the related work on how their work differs from prior work. It seems basis functions have been explored before from what I understand from their paper and also of course neural ODEs are not a new concept. So it would be very helpful to contextualize their work within existing work more clearly and delineate their contributions.*
> >
> > A: Thank you for the suggestion. We will make the suggested changes to the introduction and related work section. Indeed the idea of basis functions and neural ODEs are not new concepts.
> >
> > The contributions of our paper are highlighted at the end of the introduction, and Table 1 provides a high-level comparison. To reiterate, our contributions are as follows:
> > * We introduce a new stateful Neural ODE formulation that includes (continuous-time) Batch Normalization. Note, that our model formulation (which includes Equation 2 and 3) is different as compared to other Neural ODE formulations.
> > * We introduce a new numerical algorithm based on basis functions for solving the forward pass of Neural ODEs with normalization.
> > * We introduce a new compression methodology for Neural ODEs using basis functions.
> > * We introduce a theorem for the conditions required to learn continuous-in-depth weight functions in the discrete regime, and empirically demonstrate the theoretical result.
> > * We show an extensive set of empirical results.
> >
> > In light of these contributions, we strongly  believe that we are significantly advancing the field of ODE-based networks. Note, that we outperform previously proposed ODE-models that used different basis functions such as the Gelerking-ODE model [3].
> >
> > *Q: Additionally, I glanced over Section 4.1 in the appendix regarding the implementational details of the projection step. It seems to require various derivatives including a Hessian computation. How does the projection step scale with the model? Is that something that’s feasible for larger models and datasets? What is the practical runtime for this step? It would be good to understand that.*
> >
> > A: We discussed the asymptotic runtime in Section 3, in the paragraph “A note on complexity.” In practice, the projection step can be faster than evaluating test set accuracy. We will include wall-clock runtimes.
> >
> > We used the Hessian explanation to show the reader an easy derivation. For any set of basis functions, Equation 18 can also be written as a double integral, without relying on an automatic differentiation algorithm.
> >
> > For very large models, note that the time complexity of projection is based on the depth. E.g., for a 1,000 layer model with 1 billion parameters (whose width is 1 million parameters), projecting it to 500 layers would cost:
> >    * Calculate a 500x500 and 1000x500 matrix with numerical integration
> >    * Perform one 500x500 matrix factorization
> >     * 1 million 500x500 pre-factorized matrix solves, which can be executed in parallel on many machines by partitioning the model along its width.
> >
> > Note also that the projection step is a one time cost: the new weights are only calculated one time on the training servers. The new compressed weights are deployed to inference servers / edge devices, where the new weights are not used without needing projection.
> >
> >
> > [1] B. Chang, L. Meng, E. Haber, F. Tung, and D. Begert. Multi-level residual networks from dynamical systems view. In Proceedings of the International Conference on Learning Representations, 2018.
> >
> > [2] A. F. Queiruga, N. B. Erichson, D. Taylor, and M. W. Mahoney. Continuous-in-depth neural networks. arXiv preprint arXiv:2008.02389, 2020.
> >
> > [3] S. Massaroli, M. Poli, J. Park, A. Yamashita, and H. Asma. Dissecting neural odes. In Advances in Neural Information Processing Systems, 2020

---

> ### Comment · Reviewer_KZk9 · 2021-08-19
> **Thank you!**
>
> Thank you for the detailed responses.
>
> I spent some time going through the answers and I believe the contributions of the paper are now much more accessible. Also some of the design choices are more clear now.
>
> There is two more questions I would like to ask the authors:
>
> 1. Is there a class of basis function that seems less trivial and can highlight the advantages of your contribution in an experiment? I would love to see some results.
>
> 2. Any way that the authors could improve upon the training results for multi-refinement training? As the authors mention themselves in their paper the current training method achieves lower accuracy than regular training. That seems a potentially limiting factor in practice since I assume the authors probably already spent some amount of time to tune the multi-refinement process. And that was still not enough to obtain the accuracy of other reference methods.
>
>
> I think that some interesting discussion and results in those two directions would encourage me to raise my score.
>
> However even with these additional experiments, I am still not sure whether the paper would be a clear accept for me. There is just too many updates to the manuscript before the contributions are more accessible to a broader audience. Unfortunately, the authors cannot provide an revised version of their paper at this point.

---

> > ### Author Response · Authors · 2021-08-28
> > **Thank you for considering our responses.**
> >
> > We thank the reviewer for taking the time to read our responses, and we are glad that we were able to address the concerns. We thank the reviewer for the additional constructive feedback, to which we reply below:
> >
> > *Q: Is there a class of basis function that seems less trivial and can highlight the advantages of your contribution in an experiment? I would love to see some results.*
> >
> > The purpose of the paper was not necessarily to show which basis functions worked best, but instead to show what a basis-function framework enables. Our framework and code extends to any set of basis functions, but we discussed only three which allowed us to demonstrate the proposed Stateful-ODE and proposed compression method.
> >
> > We again thank the reviewer for recommending for us to include the rationale for our selection. However, we do not want to make a definitive statement of “Basis functions #1 are better than basis functions #2.” We fear that including a “less trivial” basis function class that performs poorly in our experiments would give the reader an impression that we do not want to make. We leave open the possibility that other classes will perform better in some situations; we expect it to be problem-specific.
> >
> > We would also like to push back on the idea that these are “trivial” basis functions. Yes, they are the first-order versions, but the function classes themselves are not trivial to implement in this deep learning context. (We needed to use a cutting-edge library, JAX.) We did not think that repeating experiments for higher-order versions (e.g. piecewise quadratic) would enhance the presentation of the contributions of the paper.
> >
> >
> > *Q: Any way that the authors could improve upon the training results for multi-refinement training? As the authors mention themselves in their paper the current training method achieves lower accuracy than regular training. That seems a potentially limiting factor in practice since I assume the authors probably already spent some amount of time to tune the multi-refinement process. And that was still not enough to obtain the accuracy of other reference methods.*
> >
> > We actually did not spend a lot of time tuning this method. We followed the guidelines proposed in [1] and [2]. We will add a line to the results section where we describe the simple heuristic: We evenly space out the refinement epochs before the first learning rate decay epoch. (Note that we also do not discuss the learning rate schedule, and simply apply others’ knowledge.)
> >
> > For now, we accept that this introduces regularization that will decrease the peak-accuracy of the model. The previous papers [1] and [2] argue that the sacrifice of accuracy buys training speed-up. We further show that the sacrifice of accuracy buys the advantage of compressibility.
> >
> > We agree that studying the multi-level refinement algorithm in more detail is merited, since there is certainly room for improvement. We did not do any hyper-parameter sweeps or try to improve the algorithm in any way for this paper. We highlight that the basis-function interpretation generalizes the multi-level refinement algorithm, and future work might take advantage of this to improve it. However, we think improving the algorithm is out-of-scope of this paper, and instead merits a dedicated paper.
> >
> > 1] B. Chang, L. Meng, E. Haber, F. Tung, and D. Begert. Multi-level residual networks from dynamical systems view. In Proceedings of the International Conference on Learning Representations, 2018.
> > [2] A. F. Queiruga, N. B. Erichson, D. Taylor, and M. W. Mahoney. Continuous-in-depth neural networks. arXiv preprint arXiv:2008.02389, 2020.

---

> > ### Author Response · Authors · 2021-08-31
> > **Feedback**
> >
> > We would like to thank you again for the very constructive feedback and we hope that our response was helpful, and addressed your concerns. We strongly believe that we present novel innovations that are relevant and of interest for the community. We assure that we are able to address all of your concerns (and those of the other reviewers) in the camera ready version of the paper, if we are given the opportunity. Please let us know if we can address any other comments. We would be glad to share a more extended discussion during the remaining time.

---

> > > ### Author Response · Authors · 2021-09-03
> > > **Thanks**
> > >
> > > We would be thankful if the reviewer has time to revisit our latest response and reconsider his score before the discussion period ends.

---

> ### Comment · Reviewer_KZk9 · 2021-09-07
> **Update to my review**
>
> Dear Authors,
>
> apologies for my slow response. I think that my overall assessment is largely remaining the same. It's an interesting contribution but not really a clear accept for me at this point. It also seems that other reviewers raised similar issues.
>
> To summarize, I would love to see the contributions scoped out in a more accessible manner as well as the actual technical details of the proposed architecture that are currently missing -- especially given the departure from exiting work in the literature (which is good but should be clearly explained and motivated).
>
> Nonetheless, the authors put a reasonable effort into their responses during the discussion phase and I feel positive about the changes the authors proposed. I will raise my score to 6 hence.
>
> Finally, I would just like to re-emphasize that I don't agree with the authors' assessment that more complex basis functions are irrelevant to show the contributions of their proposed architecture. After all, there is a huge overhead as mentioned by the authors themselves in order to have a working implementation that is fully compatible with their proposed approach. Consequently, I think it is important to see how coming up with really interesting - maybe even domain-specific basis functions - can help improve the performance and thus _justify_ the overhead of the model and its implementation.

---

> > ### Author Response · Authors · 2021-09-07
> > **Thank you for changing your score**
> >
> > We would like to thank you for the great discussion and for your consideration to change your score to 6.
> >
> > We agree with your suggestions for further exploring basis functions -- our only concern is that we are not able to include so many directions, while also addressing the more important concerns of devoting more space within the page limits to more explanations of the fundamental concepts that we introduced. We think constructing domain-specific basis functions and associated algorithms will be an excellent follow up paper, and is a direction which we will pursue.
> >
> > We think our results regarding CT-BatchNorm and compression as they stand (considering three basis functions) are enough to justify this approach.

---

### Official Review · Reviewer_FQch · 2021-07-16

**Rating:** 6
**Confidence:** 4

**Summary:**

The paper proposes weight compression technique for learnt ODE-Nets without retraining the model while maintaining near state-of-the-art performance. Their proposal includes treating the weight matrices as the linear combination of continuous-in-time basis functions, where the linear combination coefficients are learnt by optimizing the loss function between the learnt weights and the approximate weights (computed by combining basis functions). Empirical evidence suggests that such a scheme compresses ODE-Nets on image classification and sentence-tagging tasks.

**Main Review:**


Strengths:
---------------

- Compression of the weight matrices is done post training and does not require re-training or any feed-forward passes through training data.


Weakness:
---------------

- Novelty of the compression scheme seems incremental.

- Model compression using the proposed scheme only guarantees drop in model size. Paper lacks computational cost of performing inference ( Algorithm~1 suggests that the inference cost will be higher than forward pass of the uncompressed model ).

Questions for Authors:
---------------

- It would have been better to showcase the inference time comparison between the compressed and uncompressed scheme. If for each input, the weights are constructed on the fly with the basis function, this additional cost is non-trivial and needs to be accounted for. Instead if the weights are simply constructed from the compressed model, then the inference proceeds as in the uncompressed scheme, then we did not gain anything from the computational standpoint.


- Could authors elaborate a bit as to why is it hard to incorporate BatchNorm as one of the layers in the general function $f$ for the ODE $\frac{dx}{dt} = f(x)$?

- Is there any ablative study as to what benefits do you get by incorporating CT-Batch Norm in your proposed scheme?

- Do you count the basis function parameters as well as linear coefficients in the number of parameters shown in the tables?

- Did you run any of the baselines learning schemes with the architectures used in the experiments section? Its not clear from the paper if the baselines used the same architecture design as ContinuousNetV2 (barring the continuous time batch-norm).

Writing Clarity:
---------------

- Line 33: Clearly state what are gradient parameters and how they differ from state parameters.

- Missing related works

	ODE/PDEs:
	Continuous Time RNN (by Rosenblatt),
	Neural Controlled Differential Equations for Irregular Time Series (https://proceedings.neurips.cc//paper/2020/file/4a5876b450b45371f6cfe5047ac8cd45-Paper.pdf),
	Incremental RNNs (https://openreview.net/forum?id=HylpqA4FwS),
	NeuPDE ( http://proceedings.mlr.press/v107/sun20a.html),
	Time Adaptive RNNs (https://openaccess.thecvf.com/content/CVPR2021/papers/Kag_Time_Adaptive_Recurrent_Neural_Network_CVPR_2021_paper.pdf)

	Related works on the deep network compression is missing.


**Time Spent Reviewing:**

5

---

> ### Author Response · Authors · 2021-08-11
> **Response to Reviewer FQch**
>
> We thank the reviewer for the very detailed and thoughtful response and are glad to hear that the reviewer considers our key contribution, a posteriori compression of the weight matrices, as **strength**.
>
>
> *Q: Novelty of the compression scheme seems incremental.*
>
> A: Could you please elaborate on this comment, and explain what you mean by incremental?
> We are puzzled by this comment since we are not aware of any scheme that is closely related to what we have proposed in the context of ODE-based networks.
>
>
> *Q: Model compression using the proposed scheme only guarantees drop in model size. Paper lacks computational cost of performing inference ( Algorithm~1 suggests that the inference cost will be higher than forward pass of the uncompressed model ).
> It would have been better to showcase the inference time comparison between the compressed and uncompressed scheme. If for each input, the weights are constructed on the fly with the basis function, this additional cost is non-trivial and needs to be accounted for. Instead if the weights are simply constructed from the compressed model, then the inference proceeds as in the uncompressed scheme, then we did not gain anything from the computational standpoint.*
>
> A: We agree that inference times are useful: we neglected to add them to the Tables. We will add additional columns to the table. For instance, the inference times over the entire test set for Table 3 are as follows:
>
> | Model                   | Inference Time (s) | Speedup |
> | ----------------------- | ------------------ |---------|
> | ContinuousNetV2 (ours)   | 3.5                |  -      |
> | (compressed)             | 2.3               | 1.5     |
>
>
> There are multiple factors that contribute to improving inference time:
>
> * The major place cost savings come from is by changing the time step size, which reduces the number of steps. Model inference time is directly proportional to the number of time steps: inference becomes faster as NT decreases. (That is, flop count is directly proportional to NT.) As shown in Figures 4b and 5b, the projection step helps improve the robustness of changing NT, allowing for more extreme FLOP-reduction. We can include these runtimes as well, and we will discuss the practical FLOP-count implication in the discussion of these figures.
>
> * Further, cache misses can dominate runtime. Even without changing NT, the memory footprint for the calculation is lower, and less data needs to be fetched from RAM or sent over the network. For example, compressing a 16GB model down to 8GB model would result in less memory movement during the forward pass. The cost of a few extra FLOPs needed to evaluate the basis functions after loading the coefficients into cache from memory is lower than the penalty of twice as many cache misses.
>
> * Decreasing memory footprint also affects multi-tenancy systems (multiple models sharing an accelerator). For example, if you have a GPU with only 16GB, one can fit two 8GB models into GPU RAM, which provides cost savings in real-world systems. Likewise, the same principal applies for very large models that require multiple accelerators for inference; compressing the model results in needing fewer accelerators, which also yields resource consumption gains.
>
>
>
> *Q: Could authors elaborate a bit as to why is it hard to incorporate BatchNorm as one of the layers in the general function f for the ODE dx = f(x)?*
>
> A: See the difficulties discussed by [1], and [2].
>
> The difficulty is that BatchNorms are not input-output functions. The training step of a batchnorm layer has a side effect: traditionally, the BatchNorm “class” has a “self.mean += average(x)” step inside of it, that is not visible to the output or the gradient optimization algorithm. The formula would look like:
>
>     dx/dt = BatchNorm.forward(x)
>     BatchNorm.mean = (1-alpha)*BatchNorm.mean + alpha*batch_average({x1,x2,x3...})
>
> The second equation does not fit into previous Neural ODE formulations. Each BatchNorm evaluation has its own internal state: e.g. suppose an 18 layer ResNet has 18 different BatchNorms: traditionally, there are 18 seperate BatchNorm classes with 18 internal variables. The existing ODE formulations require pure-functional interpretations of the ResNets, and the mathematical formulation does allow for this internal mechanism. Thus, existing Neural ODE formulations have no way to turn those 18 BatchNorms into 1 continuous BatchNorm.
>
> One of our main contributions is introducing a new Neural ODE formulation that includes a continuous notion of “internal states”, with a pure-functional implementation. We strongly feel that this is a significant contribution that is useful for the broader ODE-net community.
>
> *Q: Is there any ablative study as to what benefits do you get by incorporating CT-Batch Norm in your proposed scheme?
> A: In our development, getting the CT-BatchNorm to work properly was necessary to get SOTA performance. Other literature discussed the importance of BatchNorms for ODE-Nets [1], [2]. We do agree with the reviewers that an ablation study would be a useful addition. We can add accuracy results from our earlier prototypes, which include a) no BatchNorm, b) a naive discrete BatchNorm, which is equal to constant-in-time basis functions, and c) CT-ReZero instead of BatchNorm.*
>
> *Q: Do you count the basis function parameters as well as linear coefficients in the number of parameters shown in the tables?*
>
> A : The parameter numbers given in the tables counted every individual parameter used in the models: the discrete layers, plus every parameter in each linear-coefficient to each of the basis functions of the continuous layers. That is, the parameter count is an exact representation of the memory footprint (i.e., number of floats).
>
> *Q: Did you run any of the baselines learning schemes with the architectures used in the experiments section? Its not clear from the paper if the baselines used the same architecture design as ContinuousNetV2 (barring the continuous time batch-norm).*
>
> A: The baselines we marked in Figures 4 and 5 were the uncompressed models. The purpose of the baseline was to show that the trained model can achieve the similar performance after deriving new models by compressing its weights and shortening its graph.
>
>
> *Q: Related works.*
>
> A: Thanks for pointing out these references. We are happy to extend the related work section and cite the suggested papers. We will also add a paragraph about distillation and quantization for compressing deep neural networks.
>
> [1] J. Gusak, L. Markeeva, T. Daulbaev, A. Katrutsa, A. Cichocki, and I. Oseledets. Towards understanding normalization in neural ODEs. In ICLR 2020 Workshop on Integration of Deep Neural Models and Differential Equations, 2020.
>
> [2] W. Xu, R. T. Chen, X. Li, and D. Duvenaud. Infinitely deep bayesian neural networks with stochastic differential equations. arXiv preprint arXiv:2102.06559, 2021.

---

> > ### Comment · Reviewer_FQch · 2021-08-31
> > **Response to Author Rebuttal**
> >
> > Thank you for answering most of my questions.
> >
> > My comment regarding the incremental nature of the compression scheme is to simply indicate that the compression idea based on basis function is available in the literature and the paper lacks discussion on the compression schemes in the related works (there are many complex compression schemes available in the neural network literature: for ex. (a) Tensor Decomposition for Compressing Recurrent
> > Neural Network https://arxiv.org/pdf/1802.10410.pdf, (b) Tensorizing Neural Networks https://papers.nips.cc/paper/2015/file/6855456e2fe46a9d49d3d3af4f57443d-Paper.pdf ).
> >
> > Thank you for clarifying the need for continous time batch norm. I would suggest you to expand on this explanation in the updated version.
> >
> > Since CT-BatchNorm is indeed a major contribution of this paper, it would have seriously helped if the authors provided ablative studies around the presence/absence of this scheme and how much benefit do you actually get from using the proposed batch norm scheme.
> >
> > Thank for you showing some results for inference time comparison which indicates that the compression scheme is better than the uncompressed variant. It would be a good idea to include inference time as well as the FLOPs required for inference in the updated version of this work.
> >
> > I agree with other reviewers in the restructuring and simplyfing the paper in terms of notation and language. It would be hard to understand for researchers without Neural ODE background.
> >
> > Since some of the author responses helped clarify my comments regarding the proposed scheme, I am increasing my score to 6.

---

> > > ### Author Response · Authors · 2021-08-31
> > > **Thank you very much**
> > >
> > > We would like to thank you very much for your time, the fruitful discussion and your detailed and constructive feedback. We are very grateful that you changed your score to 6.
> > >
> > > We will discuss the suggested references in our updated related work section and discuss other compression techniques more generally. We have all the results for an ablation study at hand, and we will better highlight the great performance boost that is gained by our proposed  continuous time batch norm layer. We have also augmented the tables with inference times, and we will add FLOPs in addition.

---

> ### Author Response · Authors · 2021-08-30
> **Feedback**
>
> We hope that our response was helpful, and addressed your concerns. Please let us know if we can address any other comments. We would be glad to share a more extended discussion during the remaining time.

---

### Official Review · Reviewer_1fyo · 2021-07-18

**Rating:** 6
**Confidence:** 4

**Summary:**

This paper discusses using piecewise constant or piecewise linear basis functions to represent time-dependent parameters in an ODE-net. This can also be used to represent parameters in a time-dependent BatchNorm layer. A change of basis can be performed to reduce the number of parameters or increase the discretization step sizes.

**Limitations And Societal Impact:**

The scope is limited to discretized networks that are simply expected to behave like ODE solutions. This expectation is a small leap of faith and should be discussed more in the paper. It’d be more convincing to see experiments making use of adaptive step size solvers and/or applications involving strong reliance on accurate ODE solutions. The batch normalization method should be ablated against as it's not clear if this leads to faster training or performance improvements. Furthermore, the numerics involved with solving an ODE with potentially very large or unbounded Lipschitz is not discussed sufficiently.

**Main Review:**

Overall, this paper uses concepts from continuous-time/depth neural networks to improve their discretized variants, e.g. a ResNet. I like that the paper shows models with higher-order discretizations and continuous basis functions behave more similarly to continuous-depth neural networks, whereas the piecewise-constant (ResNet) counterpart behaves worse, especially during basis function compression.

However, the scope of the paper seems to be limited to learning simple mappings from input to output, and does not discuss applications that ODEs are more useful at, e.g. irregularly-sampled time series prediction and continuous normalizing flows. These applications either require more complex trajectories or directly make use of the right-hand-side of ODEs, and both benefit significantly from adaptive step size solvers in practice.

With the use of fixed step size solvers, it’s not clear whether the discretized model is still an approximation to the true ODE solution. The observation that piecewise-constant basis functions do not compress well may simply be due to them not being a close approximation of an ODE. It’d be more convincing to see experiments on the aforementioned applications.

Another concern with the use of piecewise-constant/linear basis functions is the lack of smoothness for higher-order solvers. When using a method such as RK4, the step sizes must correspond to the basis function’s Delta t. This again goes against the use of adaptive step sizes. This property makes it difficult to perform long-range predictions, e.g. for irregularly-sampled time series predictions. Why not explore more complex or smoother basis functions?

Regarding continuous-time batch normalization, it doesn’t seem clear to me that using this is beneficial in practice. While the paper discusses efficiently storing a time-dependent mean/var/bias/scale, it’s not clear if this actually improves convergence during training. Normalization seems like a weird operation to have in an ODE since it forces the model to predict outputs based on relative (and stochastic due to mini-batching) differences instead of absolute position. This dependence makes its interpretation as a dynamical system following a vector field less intuitive. In any case, it would be more convincing if the authors can run ablation experiments to discuss the merits of using continuous-time batch normalization.

Furthermore, the lack of a Lipschitz bound for normalization is concerning. If a hidden unit is the same value (e.g. zero) for all samples in a mini-batch, then there is a division by zero. Likewise, it was shown that multihead attention layers are not Lipschitz [1]. Though I suspect these aren’t major concerns unless one tries to solve the ODEs extremely accurately.

[1] “The Lipschitz Constant of Self-Attention” Kim et al.

----

Update after author response: Thanks for the clarifications. I generally agree with the response; however, the paper likely should be rewritten to make it clear that this is regarding discrete-time networks and to re-iterate on the scope and expectations of this work. The naming convention for continuous-time BN can also be misleading, since this is not applied in a continuous-time setting. Just my two cents, but answering these questions may be helpful: why is compressing an ODE-net an important task? What if you simply said you're compressing ResNets (and mention extending to networks that are parameterized by more sophisticated ODE solver schemes)? This may help increase the number of readers while also clarifying scope.

**Time Spent Reviewing:**

3

---

> ### Author Response · Authors · 2021-08-11
> **Response to Reviewer 1fyo**
>
> While we value the comments of Reviewer 1fyo very much and do think the review poses very interesting thoughts, we feel that the assessment simply misses the whole point of our work: we do not try to learn an ODE. The concept of ODE-Nets is just a useful construct, and we introduce new algorithms that can take advantage of this, specifically for input-output models. We feel that the criticisms do not apply to our work. Also, the suggested other problems (continuous normalizing flows, and irregularly sampled time series data) are orthogonal to our problems and out of the scope of our work.
>
> We choose our benchmark problems specifically to compare to the results from recently published ODE-type models at NeurIPS, ICLR, etc. Indeed, problems such as CIFAR10, MNIST, and Part-of-Speech tagging are standard benchmark problems in the represent which represent a specific class of industry-relevant tasks.
>
> In light of this, we would like to kindly ask the reviewer to reassess the score for our paper.
>
> *Q: However, the scope of the paper seems to be limited to learning simple mappings from input to output, and does not discuss applications that ODEs are more useful at, e.g. irregularly-sampled time series prediction and continuous normalizing flows. These applications either require more complex trajectories or directly make use of the right-hand-side of ODEs, and both benefit significantly from adaptive step size solvers in practice.*
>
> A: Both Continuous Normalizing Flows and irregular-sampled time-series are orthogonal problems requiring separate discussions. We agree with the reviewer that time series problems are excellent applications for different ODE-inspired networks such as continuous-time recurrent neural networks, and we do have other papers that solely focus on time series problems.
>
> *Q: With the use of fixed step size solvers, it’s not clear whether the discretized model is still an approximation to the true ODE solution. The observation that piecewise-constant basis functions do not compress well may simply be due to them not being a close approximation of an ODE. It’d be more convincing to see experiments on the aforementioned applications.*
>
> A: We concur that verifying that the discrete neural network learns an ODE internally is indeed an important concern. Methods for verifying this were discussed in [1] and [2]. In the submitted paper, Figures 4b and 5b show that the models are able to generalize to different numbers of time steps (NT), which validates that the internal depth-wise dynamics of the model are a close approximation to some ODE. We could expand the results section to discuss how to validate ODE approximation, but we think it has been sufficiently discussed in the literature.
>
> Note that model compression (changing the number of basis functions) is different than changing the number of time steps. Indeed, closeness to the ODE solution plays a key role in compressibility but not necessarily in the way one might expect; this is part of the content of Theorem 1. A more accurate approximation of the ODE (e.g. step size -> 0) may not yield better compression.
>
> In fact, our results show that piecewise constant functions do compress well (see Figures 4a and 5a). Our experiments in Figure 3 showed that the Euler integrator does not compress well in general, which is the subject of Theorem 1.
>
>
> *Q: Another concern with the use of piecewise-constant/linear basis functions is the lack of smoothness for higher-order solvers. When using a method such as RK4, the step sizes must correspond to the basis function’s Delta t. This again goes against the use of adaptive step sizes. This property makes it difficult to perform long-range predictions, e.g. for irregularly-sampled time series predictions. Why not explore more complex or smoother basis functions?*
>
> A: We disagree very much with this assessment. Firstly: the time step size and basis function element size can be different; in Figures 4 and 5 we change them independently. Secondly, as the reviewer summarized, in our work, we have specifically focused on analyzing the discrete regime, both theoretically and empirically.
>
> The infinitely-deep limit is not practically relevant to real-world ML problems.
> That is why, in Section 5, we performed the theoretical analysis using discrete finite differences.
> We show that Lipschitz continuity is sufficient but not necessary. Instead, our theoretical analysis shows that the necessary condition is instead smoothness at the scale of the discrete time steps. That is, for discrete models, the curvature over the span of Delta T is more important than Lipschitz continuity in the infinitely-deep (infinitely small Delta T) limit.
>
>
> *Q: Regarding continuous-time batch normalization, it doesn’t seem clear to me that using this is beneficial in practice. While the paper discusses efficiently storing a time-dependent mean/var/bias/scale, it’s not clear if this actually improves convergence during training. Normalization seems like a weird operation to have in an ODE since it forces the model to predict outputs based on relative (and stochastic due to mini-batching) differences instead of absolute position. This dependence makes its interpretation as a dynamical system following a vector field less intuitive. In any case, it would be more convincing if the authors can run ablation experiments to discuss the merits of using continuous-time batch normalization.*
>
> A: We are very surprised that the reviewer thinks that batch normalization is not beneficial in practice. Indeed almost all state-of-the-art neural networks for computer vision tasks heavily rely on the `magic’ of batch normalization layers.
> In Tables 2 and 3, we compare the performance to other ODE-based networks and we clearly show that our proposed model with continuous-time batch normalization outperforms these other models that do not use ODE-compatible batch normalization layers.  (For computer vision models, the challenge is generally to push the performance way above 90% accuracy, since it is easy to design a model that performans in the high 80s.)
> During our own development experience, getting continuous batch normalization right was the biggest challenge to achieving state-of-the-art-performance. We also tried other non-normalization methods that were easier to implement as basis functions, such as ReZero [4], but these were unable to surpass 89%. We are happy to include these “failed” prototypes in the context of an ablation study that shows the importance of CT-BatchNorm.
>
> *Q: Furthermore, the lack of a Lipschitz bound for normalization is concerning. If a hidden unit is the same value (e.g. zero) for all samples in a mini-batch, then there is a division by zero. Likewise, it was shown that multihead attention layers are not Lipschitz [1]. Though I suspect these aren’t major concerns unless one tries to solve the ODEs extremely accurately.*
>
> A: See our comment above regarding continuity. Yes, in practice, it is not a major concern for that reason.
> The division-by-zero is an interesting point, but it is not specific to ODE-Nets. General BatchNorm implementations have tricks to mitigate this threat; e.g., one can always add a small epsilon to the denominator (the variance estimate) to ensure that blowup doesn’t occur. It was not an issue in practice for our models.
>
> [1] Ott, Katharina, et al. "ResNet After All: Neural ODEs and Their Numerical Solution." International Conference on Learning Representations. 2020.
>
> [2] Queiruga, Alejandro F., et al. "Continuous-in-depth neural networks." arXiv preprint arXiv:2008.02389 (2020).
>
> [3] J. Gusak, L. Markeeva, T. Daulbaev, A. Katrutsa, A. Cichocki, and I. Oseledets. Towards understanding normalization in neural ODEs. In ICLR 2020 Workshop on Integration of Deep Neural Models and Differential Equations, 2020.
>
> ​​[4] Bachlechner, Thomas, et al. "Rezero is all you need: Fast convergence at large depth." arXiv preprint arXiv:2003.04887 (2020).

---

> ### Author Response · Authors · 2021-09-01
> **Thank you for considering our responses.**
>
> We thank the reviewer for the thoughtful feedback and considering our responses. We are grateful that the reviewer increased their score to a 6 in light of our clarifications. (We apologize for the delayed reply, as we did not receive a notification when you edited your original response.) Below we comment on individual points:
>
> *The naming convention for continuous-time BN can also be misleading, since this is not applied in a continuous-time setting.*
>
> Thank you for the excellent suggestion; we see how that can confuse the reader. We will rename it to “continuous-depth” BN.
>
> *Just my two cents, but answering these questions may be helpful: why is compressing an ODE-net an important task? What if you simply said you're compressing ResNets (and mention extending to networks that are parameterized by more sophisticated ODE solver schemes)? This may help increase the number of readers while also clarifying scope.*
>
> This is excellent feedback, and we can adjust the problem framing in our paper with that interpretation as suggested. Our method can indeed be used to compress shorter ResNets, but there is a caveat: ResNets are not necessarily ODEs as shown in [1, 2]. You need to start training with the ODE-Net perspective and use a training regiment that ensures it is an ODE. As discussed by [2], one needs to train the ODE-Net with the higher-order schemes, and then the ODE-Net can generate a ResNet after training (but not necessarily vice versa.) The ODE-Nets we trained in our paper generated identical graphs as ResNets, but their weights have additional properties that enable compression.
>
> However, we do believe that compressing ODE-Nets is in-of-itself an important task, given the active state of research into ODE-Nets as their own class of networks specifically. We think that our paper is a “killer application” that will bring ODE-Nets to be useful to the greater number of readers you suggest.
>
> [1] Ott, Katharina, et al. "ResNet After All: Neural ODEs and Their Numerical Solution." International Conference on Learning Representations. 2020.
> [2] A. F. Queiruga, N. B. Erichson, D. Taylor, and M. W. Mahoney. Continuous-in-depth neural networks. arXiv preprint arXiv:2008.02389, 2020.

---

### Author Response · Authors · 2021-08-11
**General Comment**

We would like to thank all the reviewers for the valuable and thoughtful feedback.

All reviewers agree that this is novel work, and that we have good experiments. We agree that the presentation of the results could use some improvement, in particular when we are presenting something novel with a lot of technically important details, and we are grateful for the constructive responses. With some minor tweaks, and adding additional data points from our experiments, we believe we can fully address the concerns.

We stress that we propose a novel **stateful** neural ODE model that enables a posteriori model compression and the construction of continuous-time batch normalization. We provide a high-performance implementation in Jax and our continuous-time batch normalization layer is plug-and play, i.e., it can be easily incorporated into any other model in Jax, which wants to make use of a StatefulOdeBlock.

One of the main concerns was that we did not explicitly demonstrate the advantage of the continuous-time batch normalization layers in the form of an ablation study. Indeed, the importance of batch normalization layers is very well known in the context of computer vision tasks.  Tables 2 and 3 compare the performance of our model compared to other state-of-the-art ODE-based networks that do not use continuous-time batch normalization layers. Our model clearly outperforms all the other ODE-based models, with less or a roughly similar number of parameters. We can certainly add an additional row in all the tables to make more explicit the accuracy drops without the novel continuous-time batch normalization layer. For instance, in Table 3 and 4, our prototype models achieved at best 89% accuracy on CIFAR10 without the continuous-time batch normalization layer.

The suggestions of the reviewer greatly helped us to revise and restructure the presentation of work (e.g., we have streamlined our introduction, extended the related work section and expanded the discussion of the continuous-time batch normalization layer and better highlighted the importance of Theorem 1 for our work).

Incorporating the feedback from all reviewers, we will make the following changes to the text:
* We will restructure the introduction and extend the related work section. Further, we will use the additional page to expand our discussions of proposed ideas.

* We expand Section 5 and better highlight the importance of the Theorem for our work.

* We add an additional column to the tables that show the reduction of inference time.

* We add additional rows to the tables that show the performance of our model without continuous-time batch normalization layers.

---

### Decision · Program_Chairs · 2021-09-27

**Decision:**

Accept (Poster)

**Comment:**

This paper presents a compressed representation of ODE-based neural networks, and analyzes its computational and implicit regularization properties. Experiments cover a good range of architectures, including both convolutional and transformer-based.

The reviewers believe there are worthwhile ideas in the paper and haven't identified anything that looks like a critical flaw. They raised some concerns about clarity and identified numerous ways in which the work could be extended. Reviewers also expressed skepticism of the usefulness, either because compression might not translate into practical gains, or because this architecture might lose various benefits associated with neural ODEs.

My own impression is that the submission is pretty well written by the standards of a NeurIPS paper, and the methods are described clearly even if the motivation could sometimes be made more explicit. The reviewers' suggestions seem to me (as a non-expert on this topic) to be sensible ones which would indeed improve the paper, but any paper that tried to do all of them would be unreasonably expansive. Regarding significance, the proposed architecture seems like a natural one to explore, and a paper which does it carefully (as this one does) is making a worthwhile contribution. Therefore, I recommend acceptance despite the slightly-low scores.